# Development and Fuzzy Logic-Based Optimization of Golden Milk Formulations Using RW-Dried Turmeric Powder: A Study on Shelf Life, Sensory Attributes, and Functional Properties

**DOI:** 10.3390/foods14172948

**Published:** 2025-08-24

**Authors:** Preetisagar Talukdar, Kamal Narayan Baruah, Pankaj Jyoti Barman, Shagufta Rizwana, Sonu Sharma, Ramagopal V. S. Uppaluri

**Affiliations:** 1Department of Chemical Engineering, Indian Institute of Technology, Guwahati 781039, Assam, India; preetisagar1891@gmail.com (P.T.); ramgopalu@iitg.ac.in (R.V.S.U.); 2Department of Food Technology, Assam Royal Global University, Guwahati 781035, Assam, India; srizwana@rgu.ac; 3Department of Biotechnology, Sharda School of Bioscience & Technology, Greater Noida 201310, Uttar Pradesh, India; 4Department of Community Medicine, Dhubri Medical College and Hospital, Dhubri 783325, Assam, India; pankajbarman6411@gmail.com; 5Research & Development Department, Cambridge Treats Inc., 115 Goddard Cres, Cambridge, ON N3E 0B1, Canada; sonucanada45@gmail.com

**Keywords:** refractance window drying, turmeric, sorption isotherm, shelf-life, storage study

## Abstract

The storage characteristics of folic acid and NaFeEDTA fortified in a refractance window-dried turmeric powder base and its subsequent application to the formulation of nutritionally functionalized golden milk have not been addressed in previous studies. Golden milk is a staple food and ideal matrix for the fortification of important nutrients such as iron and folic acid. With this motivation, the present study assesses refractance window (RW)-dried turmeric powder fortified with folic acid and NaFeEDTA in terms of its moisture isotherm, permeability of packing material, and storage parameters to calculate its shelf life. Further, a sensory analysis was conducted based on the fuzzy logic method to obtain the best constitution of RW-dried turmeric powder in milk. For the best formulation of golden milk, the characteristics of the product under unrefrigerated and refrigerated conditions were evaluated in addition to the storage study. Additionally, moisture content (MC), total flavonoid content (TFC), total phenolic content (TPC), antioxidant activity (AA), curcumin content (CC), color indices, bulk densities, solubility, swelling power, and water binding capacities were studied with respect to time. The results demonstrated a healthy shelf life of 184, 187, and 183 days for RW-dried, folic acid-fortified, and NaFeEDTA-fortified RW-dried turmeric powder samples, respectively, in the zipper pouch system. The fuzzy scores ranked the sample with 1 g concentration of turmeric powder as the best, considering taste, aroma, mouthfeel, aftertaste, consistency, and overall acceptability. The TPC, TFC, AA, and CC values for RW-dried turmeric powder in milk were 876.21 mg GAE/100 mL, 784.61 mg quercetin/100 mL, 24.50% and 4.20% *w*/*w*, respectively. Marginal alterations were found for the RW-dried fortified and unfortified turmeric samples. This fortified golden milk has the potential for use as a health drink.

## 1. Introduction

Understanding stability and shelf life of food products is essential to ensure their quality, safety, and efficacy over time. Food storage studies are important, as they help in evaluating the effects of environmental factors like temperature, light, and humidity on food products [1]. Using food storage assays, the influence of packaging on the microbiological, chemical, and physical attributes of food products can be determined. The determination of shelf life helps to estimate the duration for which the food product remains viable for consumption. Studies are essential in the food industry, as food product quality impacts consumer satisfaction and regulatory guidelines. Based on accurate shelf-life prediction and good storage conditions, food manufacturers can reduce waste, minimize cost, and improve product performance. Shelf-life studies provide insight into the desired characteristics of a food material in terms of its sensory, chemical, physical, and microbiological properties. Compliance of the product with the label’s declaration of nutritional data throughout storage duration is necessary for consumer acceptability [2].

Food preservation methods target moisture control in processed dry foods to a significant extent. For dry food products, moisture content-related shelf-life studies target the determination of equilibrium moisture content (EMC) and water activity (a_w_) in terms of a moisture sorption isotherm. The isotherm models reported in the literature have been categorized into various kinetic models such as absorbed water monolayer (Brunauer–Emmett–Teller model), multi-layer and condensed film (Guggenheim–Anderson–de Boer model), semi-empirical (Halsey model), and purely empirical (Oswin and Smith models) models [3]. The best fitness of these models are addressed experimentally. Additionally, the influence of temperature, humidity, oxygen, and light on the characteristics of stored food products needs to be determined. The quality parameters of turmeric powder—an excellent source of curcumin with many medicinal benefits—remain a subject of current research [4,5]. Turmeric is commonly used in preparing golden milk formulations and is commonly consumed across various parts of the world, specifically in India, where it is also known as *Haldhi Dudh.* Due to its widespread consumption and being considered a staple food, golden milk is an ideal food, with folic acid and iron fortification, intended to reach a wider population. Previous studies have confirmed its efficiency and enhanced bioavailability properties [6,7].

From an Indian environmental perspective (high relative humidity and temperature of 90% and 38 °C, respectively), the accelerated storage study methodology suggested in a recent study is appropriate for fostering quicker insights into storage time relationships [8]. Previous studies applied a static gravimetric approach for distinct food products such as mango powder, garlic, etc. [9,10]. Additionally, a recent study targeted the time-dependent storage analysis of carrot powder to measure changes in moisture and protein content [11]. The results indicated that there was an increase in moisture content and a decrease in protein content over time. Storage conditions, i.e., temperature and a controlled environment, are essential parameters that are needed to enhance the shelf life of a food product. Research has been conducted to analyze the storage of various food products dried using refractance window drying (RWD) technology. For instance, RWD mango flakes were maintained in a sealed package at 35 °C after nitrogen flushing for a storage study [12]. A type III sigmoidal curve was obtained, which indicated higher and lower moisture content adsorption capacities above and below 0.5 a_w_, respectively. Furthermore, RWD acai juice powder had a shelf life of 3 months at 22 °C [1]. Similarly, RWD pomegranate leather had a shelf life of 4 and 2 months at 22 °C and 35 °C, respectively [13]. Compared to freeze drying (35.77 mg/100 g), blueberries dried using RWD (38.09 mg/100 g) exhibited higher vitamin C concentration during storage [14]. In another study, kefir powder from RWD had a shelf life of 90 days when stored at 4 °C [13]. These studies confirm the efficacy of RW-dried products compared to other drying methods in terms of quality and shelf life.

Food powder mixes are usually sensitive to storage environment factors like moisture, light exposure, oxygen, and heat [15,16]. These environmental variations affect the shelf life of food powders [17,18]. The extent of environmental influence on food powder is contingent upon its chemical composition and manufacturing processes. Usually, the shelf life of food powders depends on physical and chemical interactions, which are controlled by the water activity of the food powders [19].

RWD is a novel, less explored rapid drying process, which retains the nutritional properties of food products. Freeze drying has long been popular for preserving nutrition in food materials; however, it is high-cost, so RWD represents a promising alternative. Advantages of RWD over freeze drying include that it is considerably less expensive and has less cumbersome scalability, while providing a comparable yield [17,20]. Thus, RWD reduces the operational and transportation costs [21]. A prior study found that mint leaves had higher antioxidant activity and total phenolic and flavonoid content when dried at 70 °C using the RWD process [22]. Similarly, another study reported an advantage of RWD in that it reduced the water activity of peach slices by 40% with minimal change in color indices [23].

Fuzzy logic is a rule-based system that operates using IF–THEN statements to handle uncertainty in decision-making. Comprising a fuzzy rule base, fuzzification, inference, and defuzzification, this approach effectively translates numerical data into linguistic terms [14,24]. Therefore, it is widely used for quality evaluation, attribute analysis, and assessing the sensory characteristics of solid, semi-solid, and liquid food products [25]. One study used fuzzy logic to grade tomatoes based on their color [26]. The study found that the fuzzy controller was correctly able to grade tomatoes, matching the consumers’ perception of ripe tomatoes. In another study, characteristics of grapefruit, like taste, pH, and firmness, were determined using fuzzy logic [24]. The fuzzy controller indicated that taste and pH increased over time while hardness decreased. Another study conducted a sensory evaluation of black carrot juice based on fuzzy logic, which indicated higher sensory attributes like taste, aroma, and color for optimal formulation [18]. Sensory evaluations based on fuzzy logic analysis showed that the nutritional bar with 10 g/100 g tilapia protein hydrolysate had the highest score for overall acceptability [27]. Studies support the use of fuzzy logic-based sensory analysis for evaluating the best formulations of different food products.

To date, RW-dried, folic acid-fortified, and NaFeEDTA-fortified turmeric powder samples have not been studied in terms of assessing their moisture isotherm, shelf life, and storage, along with other associated characteristics. Considering the above-mentioned lacunae, this article addresses an isotherm and storage study of RW-dried, folic acid-fortified, and NaFeEDTA-fortified turmeric powder.

## 2. Materials and Methods 

### 2.1. Raw Materials, Chemicals, and Sample Preparation

Turmeric was procured from the market complex, Indian Institute of Technology Guwahati, Kamrup, Assam, India, and was packed in a polythene pouch to prevent contamination during its transportation. Sodium ferric ethylenediaminetetraacetate (NaFeEDTA), folic acid, potassium bromide, enzymes, and other chemicals were obtained from Sigma Aldrich, Bangalore, Karnataka, India. Subsequently, the procured raw turmeric was washed with tap water to remove surface contaminants and dirt. Then, the sample was wiped with tissue paper to remove excess water before peeling the wiped turmeric. Finally, using an adjustable slicer, the peeled turmeric was sliced into sample pieces of 1 mm thickness.

### 2.2. Refractance Window Drying

The refractance window drying (RWD) experiments were conducted using the response surface methodology (RSM) for a 1 mm turmeric slice sample. The experimental design was based on the Box Behnken Design (BBD) methodology using Design Expert software (Version 7). The BBD-based RSM design indicated 15 experimental data sets, in which three sets correspond to central point investigations. The BBD enabled better precision in the centered factor space. Thus, a 3-factor 3-level BBD with three runs at the center point was adopted for RWD investigation of turmeric slice samples. Based on initial investigations, water bath temperature, air velocity, and drying time ranges were determined to be 65–95 °C, 0.5–1 m/s, and 75–360 min, respectively. After conducting the RSM experiments, the optimized water bath temperature, air velocity, and drying time were set as 95 °C, 0.75 m/s, and 75 min, respectively [7]. Thereafter, the dried samples were powdered using a dry portable electric grinder. Then, sieved samples were obtained using the 80-mesh sieve and were analyzed as possessing an average particle size of 0.177 mm [28]. It should be noted that the optimization studies were conducted with the turmeric slices, and fortification was not carried out during this stage. Thus, these investigations were conducted as part of the follow-up work from our earlier study [6].

### 2.3. Fortification with Sodium Ferric Ethylenediaminetetraacetate and Folic Acid

Considering relevant prior fortification methods, 100 g of RW dried turmeric powder was mixed with 20 mg NaFeEDTA or 20 mg folic acid to achieve iron and folic acid-fortified turmeric powder samples [6,29,30,31,32]. For both cases, dry mixing using a spatula was performed.

### 2.4. Evaluation of Sorption Characteristics

The sorption characteristics were determined for RW-dried, folic acid-fortified, and NaFeEDTA-fortified turmeric powder samples, following the procedure for sorption characteristics established by previous studies [33]. For all the samples mentioned, adsorption isotherm studies were carried out at 40 °C. First, 20 g of turmeric powder samp les were placed in petri plates and kept in eight distinct desiccators. These desiccators constituted a saturated solution of various salts that customized the achievement of relative humidity (RH, 11.2 to 79.9%) in the environment. Eventually, the desired RH condition was met in distinct desiccators. These desiccators were placed in an incubator (Make: Labtop, Mumbai, India) at 40 °C. To prevent mold growth, a dish containing 5 mL of toluene was placed in the desiccators, which facilitated an environment with a relative humidity of over 75%. Thereafter, periodical weight measurement of the samples was conducted until moisture equilibrium (constant weight) was achieved. These samples were analyzed for moisture content. The moisture sorption isotherms were prepared in terms of the equilibrium moisture content and water activity plots. The fitness of the appropriate model for experimentally determined water activity and equilibrium moisture content was ensured using the Guggenheim–Anderson–de Boer (GAB) model, expressed as:(1)$$M=\frac{M_{0}\times C\times K\times a_{w}}{\left(1-K\times a_{w})(1-K\times a_{w}+C\times K\times a_{w}\right)},$$
where M_o_ is the monolayer moisture content (db), M is the EMC (db), and C and K are constants. In the above expression, parameter (M_o_, C, K) values were determined using the Origin Pro 9 software’s non-linear curve fitting module [34].

### 2.5. Shelf-Life Characteristics

For the RW-dried, folic acid-fortified, and NaFeEDTA-fortified turmeric powder samples, zipper pouches (14 cm × 10 cm) were employed for storing 20 g of powder samples. These pouches were then kept in a desiccator at 38 ± 1 °C. The desiccators constituted a saturated solution of potassium nitrate to facilitate an environment with 90% relative humidity. Approximately eight sample pouches for each of the three cases were kept in each desiccator to ensure that the pouches were maintained in similar environmental conditions. The sample moisture was determined at a uniform interval until a moisture content of 8% was reached. Eventually, the sample shelf life was estimated using the expression:(2)$$\int d\theta =\frac{W_{s}}{P*KA}\int_{X_{i}}^{X_{c}}\frac{dX}{RH-a_{w}},$$
where θ denotes shelf-life (days), W_s_ refers to the weight of dry solids (g), P* refers to the saturated vapor pressure of water at ambient temperature (Pa), K refers to the water vapor permeability of the packaging material (kg/m^2^ day Pa), A denotes area of the package (m^2^), RH refers to the relative humidity of the environment in which the package is placed (%), a_w_ denotes the water activity of the powder, X_i_ refers to the initial moisture content (% db) and X_c_ denotes the critical moisture content (% db).

For the packaging material, water vapor permeability K (kg/m^2^ day Pa) was determined using the expression:(3)$$K=\frac{dw/d\theta_{p}}{A_{p}P*},$$
where dw/dθ_p_ is the slope of the straight line fit of the plot drawn between time and the weight of silica gel kept within the packaging material, A_p_ is the surface area of the packaging material (m^2^), and P* is the saturation vapor pressure of the water in the packaging environment temperature (38 ± 1 °C) (Pa) [10].

### 2.6. Nutritional and Physical Characteristics of Stored Dry Curcuma Longa Powder Samples

One zipper pouch (20 g) for RW-dried, folic acid-fortified, and NaFeEDTA-fortified turmeric powder samples were taken out at regular intervals for the assessment of the physical and nutritional characteristics of the stored powder. The storage study was conducted at 38 ± 1 °C, and the samples were analyzed at 0, 9, 17, and 24 weeks. The nutritional and physical variations of the powder samples were determined following methods listed in prior studies: MC, AA, TPC, TFC, CC, color indices, bulk density, solubility and swelling power, water binding capacity, dispersion time, hygroscopicity, folic acid estimation, and iron content estimation [12,13,35,36,37,38,39].

### 2.7. Sensory Analysis

The sensory analysis was conducted by preparing golden milk samples for only the RW-dried turmeric powder samples. This was because the fortified samples used non-food-grade folic acid and iron precursors. The sensory analysis first involved a variant turmeric constitution (0.5 g–2 g in 100 mL of milk (Amul Taaza) based on RW-dried turmeric powder milk product preparation). Then, the prepared liquid samples were subjected to sensory characterization using a panel of 25 experts who possessed adequate knowledge of the product characteristics and associated sensory attributes of dairy products. With their expertise in terms of their regular involvement with the sensory evaluation of such products, sensory analysis data were customized using a 9-point hedonic scale to represent color and appearance, taste, aroma, mouthfeel, aftertaste, consistency, and overall acceptability [40,41].

Fuzzy logic was used to analyze the sensory scores of the samples. The first step involved the calculation of a triplet for all the sensory scores for each attribute (using Equation (4)). The relative weightages of quality attributes were calculated (using Equation (5)), and the product of two triplets, say, for example, T_1_ and T_2_, was determined using the expression presented in Equation (6). The overall score, O_attr_, for an attribute is the sum of the products of its triplet with the relative weightages of all attributes (as shown in Equation (7)). The membership value µ for a triplet $T=\left(a,b,c\right)$ and a score x was calculated using Equation (8). The membership of a standard fuzzy scale was determined for each x in the range [0, 100], with steps for n number of samples (using Equation (9)). The similarity S between the sample values and a standard fuzzy scale was calculated using the following expression (Equation (10)). Further, the rankings were based on the maximum similarity value for each attribute (using Equation (11)).(4)$$T=\left(\frac{\sum_{i=1}^{n}S_{i}T_{0i}}{Nn},\frac{\sum_{i=1}^{n}S_{i}T_{1i}}{Nn},\frac{\sum_{i=1}^{n}S_{i}T_{2i}}{Nn}\right),$$
where $S_{i\;}$is the sensory score given by the *i*th judge; $T_{0i},\;T_{1i},\;and\;T_{2i}\;$ are the components of the triplet associated with the iii-^th^ score; N is the number of judges; and n is the number of sensory scores.(5)$$W=\left(\frac{T_{0}}{Q_{sum}},\frac{T_{1}}{Q_{sum}},\frac{T_{2}}{Q_{sum}}\right)$$
where T_0_, T_1_, and T_2_ are the components of the triplet for the attribute and $Q_{sum}=1-\sum_{j=1}^{m}T_{0j}$ is the sum of the first components of the triplets for all m attributes.(6)$$T=\left(T_{10}\times T_{20,}T_{10}\times T_{21}+T_{11}\times T_{20,}T_{10}\times T_{22}+T_{12}\times T_{20}\right)$$
where $T_{1}=\left(T_{10},T_{11},T_{12}\right)$, $T_{2}=\left(T_{20},T_{21},T_{22}\right)$.(7)$$O_{attr}=\sum_{k=1}^{m}\left(T_{attr}\times W_{k}\right)$$(8)$$\mu =\left\{\begin{array}{l} 0,if\;x<a-b\;or\;x>a+c\\ \frac{x-(a-b)}{b},if\;a-b\leq x\leq a\\ \frac{(a+c)-x}{c},if\;a<x\leq a+c \end{array}\right\}$$(9)$$\mu_{s}=\max\left(\mu \left(T,x\right),\mu \left(T,x+n\right)\right)$$(10)$$S=\frac{\sum_{i=1}^{n}F_{si}V_{i}}{\max\left(\sum_{i=1}^{n}F_{si}^{2},\sum_{i=1}^{n}V_{i}^{2}\right)}$$
where F_si_ represents the membership values of the standard fuzzy scale and V_i_ is the sample membership values.(11)$$R_{attr}=\arg_{s}\max\;S_{attr,s}$$
where S denotes standard fuzzy scales F_1_, F_2_, and F_6_.

### 2.8. Characterization of Optimal Gold Milk Formulation

For sensory analysis-based gold milk formulation, relevant characterizations were carried out in terms of AA, TPC, TFC, and CC. The adopted procedures have been briefly summarized in the following sub-sections [36,42,43].

#### 2.8.1. Total Phenolic Content

The total phenolic content was estimated using the FCR method [35]. The sample processing procedure involved mixing 1 mL of golden milk with 1 mL of prepared FCR reagent in a vortex mixer. After 5 min of mixing, 10 mL of 7% sodium carbonate solution was added to the mixture. After subsequent vortex mixing, the final solution volume was adjusted to 25 mL. Finally, the solution was kept in a dark environment for 2 h at ambient temperature. This was followed by absorbance determination for the sample at 750 nm using a UV spectrophotometer (UV-2600, Shimadzu, Singapore). Using a calibration chart prepared with gallic acid, the TPC content of the sample was expressed in terms of mg of GAE/g sample.

#### 2.8.2. Total Flavonoid Content

The AlCl_3_ method was duly followed to determine the TFC [35]. First, this involved mixing 1 mL of golden milk with 4 mL of distilled water in a vortex mixer. Thereafter, 0.3 mL of 5% NaNO_3_ solution was added, and the mixture was subjected to vortex mixing for 5 min. Eventually, 0.3 mL of 10% AlCl_3_ was added to the mixture and was vortex-mixed for 6 min. Later, 2 mL of 1 M NaOH was added to the solution and thoroughly mixed. The final solution with a volume of 10 mL was kept in a dark environment for 30 min at ambient temperature. Thereafter, absorbance was measured at 517 nm. Using a quercetin-based calibration chart, the TFC of the sample was determined in terms of the mg of quercetin equivalent/g sample.

#### 2.8.3. Antioxidant Activity

The golden milk product was subjected to total antioxidant activity evaluation using the 1,1-diphenyl-2-picryl hydrazyl (DPPH) assay method [35]. First, 2 mL of golden milk was mixed with 2 mL of DPPH stock solution (1:1). Simultaneously, a control sample was prepared using distilled water and DPPH stock solution. Thereafter, the samples were incubated for 30 min in a dark environment at room temperature. Eventually, the absorbance was analyzed at 510 nm. The % AA was determined using the expression:(12)$$A.A\;\left(\%\right)=\frac{A_{C}-A_{S}}{A_{C}}\times 100$$
where A_c_ and A_s_ correspond to the absorbance of the control and the sample, respectively.

#### 2.8.4. Curcumin Content

The curcumin content of golden milk was determined by adopting the procedure summarized in [35]. First, 1 mL of milk was placed in a test tube and boiled with a 10 mg dried turmeric sample. Thereafter, the sample was cooled to room temperature. After this, 5 mL of 95% methanol was added to the sample and sonicated for 10 min. Thereafter, 5 mL of extract was mixed with 95% methanol to achieve a solution volume of 10 mL. The solution obtained was filtered using Whatman filter paper No. 1. After filtration, 0.4 mL of filtrate was added to 5 mL of 95% methanol. The absorbance of the samples was measured at 425 nm with a UV-VIS spectrophotometer. Using the measured absorbance of the standard and turmeric processed samples, the curcumin content of the golden milk was determined using the calibration curve in terms of % curcumin content *w*/*w*.

### 2.9. Experimental Design and Statistical Analysis

The experiments followed a completely randomized design with three replicates. The data obtained were analyzed using an analysis of variance (ANOVA), with the significance level set at 0.05, using the Origin Pro 9 software [6,7].

## 3. Results and Discussion

This work is a continuation of an earlier published work, where the RW-dried, folic acid-fortified, and NaFeEDTA-fortified turmeric samples were assessed for their different characteristics.

### 3.1. Sorption Characteristics of Powder Products

The moisture sorption behavior of RW-dried, folic acid-fortified, and NaFeEDTA-fortified turmeric samples were studied at 40 °C. The obtained sorption isotherms revealed that the equilibrium moisture content increased with increasing water activity at a constant temperature (Figure 1). This is a characteristic feature of amorphous materials with a richer constitution of hydrophilic components. The observed behavior has been attributed to the hydrophilic nature of carbohydrates and constituent protein content in the RWD-dried powder samples. For the case of low and intermediate water activities (multilayer sorption region), the equilibrium moisture content increased linearly with water activity. However, in the case of high water activities (capillary condensation region), a sharp enhancement in equilibrium moisture content was observed [36]. This trend has been reported in the literature for food materials including raisins, figs, apricots and prunes [13], grapes and apples [14], tomato pulp powder [11], and orange juice powder [15]. The available literature data confirmed that the equilibrium moisture content reduced with increasing temperature in the case of constant water activity. This trend may be due to the reduction in the total number of active sites for water binding where temperature influenced physical and/or chemical changes. For enhanced temperature cases, the water molecules are activated to higher energy levels and break away from the water binding sites of the material, thereby reducing the equilibrium moisture content [16].

In the GAB equation, the term ‘M_0_’ (monolayer moisture content) corresponds to the amount of moisture absorbed in a monolayer that exists on the adsorbent surface. Thus, the term is a measure of the availability of active sorption sites. Further, the parameter ‘C’ determines the strength of binding water molecules to the primary binding sites on the sample surface. Thus, a larger C value affirms stronger bonding between the water molecules in the monolayer and the binding sites on the sorbent surface. Also, the parameter ‘K’ is a correction factor for multilayer molecules in conjunction with the bulk liquid. For the case where the K value is 1, the molecules beyond the monolayer have the same characteristics as those of pure water [40]. Table 1 summarizes the analysis results for the GAB model to represent the experimentally measured sorption data. The table presents C, K, and M_o_ R^2^ values. This is in agreement with prior studies of the sorption isotherms of tomato pulp powder and orange juice powder [44]. Similar results were also reported for freeze-dried dates [20]. A recent study reports the GAB model as the best-fitted model for baby corn powder at 40 °C [8]. Therefore, the GAB sorption model could best represent the pertinent adsorption behavior of the RW-dried, folic acid-fortified, and NaFeEDTA-fortified turmeric RW-dried powder samples.

### 3.2. Permeability of Packaging Material

Preliminary studies were carried out to assess the suitability of the packaging material (low-density polyethylene, i.e., zipper pouch) for the packaging of turmeric powder. Figure 2 depicts the time-dependent cumulative moisture gain over time by silica gel in a zipper pouch at 38 ± 2 °C and 90% RH. The slope of the straight line fit (dw/dqp) was determined as 0.000011 kg/day. Using expression (3) for permeability determination, the water vapor permeability (K) of the zipper pouch (low-density polyethylene) for a surface area value of 0.014 m^2^ and *p** value of 6980.5 Pa (saturated vapor pressure of water at 38 °C), was obtained as 1.12 × 10^−7^ kg/m^2^ day Pa [10].

### 3.3. Shelf-Life Assessment

Using Equation (2), the shelf-life parameters of RW-dried, folic acid-fortified, and NaFeEDTA-fortified turmeric powder samples packed in zipper pouches (38 ± 1 °C) were determined. The water vapor permeability of the zipper pouches was calculated using Equation (3) and was found to be 1.12 × 10^−7^ kg/m^2^ day Pa. The initial moisture content (X_i_) of RW-dried, folic acid-fortified, and NaFeEDTA-fortified turmeric powder samples was the same for all cases at 4.27% (db), respectively. After 180 days of storage, the final moisture content of the packed RW-dried, folic acid-fortified, and NaFeEDTA-fortified turmeric powder samples were 8.60, 8.70, and 8.50% (db), respectively. Thus, it is apparent that the moisture content of the powder sample gradually increased with storage time. Also, at this stage, the sample showed a caking tendency [34]. Therefore, the critical moisture content (X_c_) was taken at this stage as 8.60% (db) for all cases.

At a storage temperature of 38 °C, the water activity values corresponding to critical moisture content for RW-dried, folic acid-fortified, and NaFeEDTA-fortified turmeric powder samples were 0.49, 0.49, and 0.48, respectively. Considering the zipper pouch surface area as 0.014 m^2^ for all cases, the total solid weight (W_s_) was 0.01918, 0.01938, and 0.01938 kg for RW-dried, folic acid-fortified, and NaFeEDTA-fortified turmeric powder samples, respectively. The saturated vapor pressure of 6980.5 Pa has been used to determine the shelf life of the samples [45]. Corresponding shelf life values were 184, 187, and 183 days for RW-dried, folic acid-fortified, and NaFeEDTA-fortified turmeric powder samples, respectively, in the zipper pouch system. In a relevant study, the authors reported a shelf life of 102 days for milk millet powders, which is comparable to the results obtained in this work [46].

### 3.4. Sensory Assessment of Powder Product

#### 3.4.1. Hedonic Scale Based Sensory Analysis

As a simple marketing rule, if consumers do not like the appearance, flavor, or texture of a food product, they do not buy it. Therefore, the overall sensory experience of a product is crucial for its commercial success. Specific protocols and methods have been developed to estimate and quantify consumers’ sensory experiences. Accordingly, the risk associated with the non-acceptability of a food product can be reduced through scientific relevance in terms of sensory descriptive analysis or sensory descriptive evaluation. Appropriate sensory evaluation allows for a useful understanding of key attributes that assist in the commercial success of food products [47].

The RW-dried turmeric powder was evaluated for its sensory characteristics. However, the folic acid-fortified and NaFeEDTA-fortified turmeric powder samples were not analyzed for their sensory characteristics due to the non-food-grade status of the deployed folic acid and NaFeEDTA fortificants. The sensory analysis was conducted by a panel of judges who provided scores for color and appearance, taste, aroma, mouthfeel, aftertaste, consistency, and overall acceptability [48].

The color and appearance of a product are a prime factor for its formal acceptance or rejection during sensory analysis. The aesthetic quality of a food product is predominantly influenced by its color, providing visual inputs for flavor identification. After color and appearance, aroma is significant for the sensory evaluation of a sample and critically influences the formal acceptance or rejection of a sample by a consumer. The smell or aroma of a product influences the olfactory glands of consumers and accordingly enhances the desire of the consumers to taste a product. Thus, a product with a good aroma attracts a consumer to taste it, and a bad or strong aroma hinders the formal acceptance of the food product. Despite having good appearance and color properties, a food product is usually accepted for its good taste. The aftertaste is an attribute that lingers in the palate of the mouth after a sample’s ingestion into the mouth and reflects upon the acceptability of the product by the consumers. The consistency and mouthfeel of a product are important sensory attributes from which to infer its degree of acceptability. However, the overall acceptability parameter has the final say in the acceptability of a product by the consumer [47,48].

A sensory analysis was performed for 100 mL of milk mixed with 0 g, 0.5 g, 1 g, 1.5 g, and 2 g of RW-dried turmeric powder. Twenty-five semi-trained panelists evaluated the samples for sensory attributes—color and appearance, aroma, taste, aftertaste, consistency, mouthfeel, and overall acceptability—using the 9-point hedonic scale (Figure 3). At lower concentrations, i.e., 0 g and 0.5 g, the sensory scores for color were liked marginally. However, other attributes were moderately liked. The sensory scores for the 1 g concentration were highest, between 8 and 9 (like very much and like moderately), for all the attributes. At 2 g, the scores for mouthfeel, aftertaste, and overall acceptability suggested that the sample was disliked by panelists.

#### 3.4.2. Fuzzy Logic-Based Sensory Analysis

The study applied fuzzy logic for the sensory attributes-based evaluation of the sensitive influence of the turmeric powder concentration. The similarity values reflected the ranks being obtained as: not satisfactory—0.00 to 0.40; satisfactory—0.40–0.60, good—0.60–0.80; very good—0.80–0.85; and excellent—0.85–1.00. The quality attributes ranking from the fuzzy logic are summarized in Table 2. The similarity value was calculated based on the fuzzy logic method, and the assigned membership for a concentration of 1.5 g of the turmeric powder was very good for the color attribute (0.825). Higher similarity values indicated the superiority of the sensory scores for that sample. The highest similarity values for all the other parameters except for color were found for the 1.0 g turmeric powder concentration sample. For aroma, while the similarity values indicated a very good rank, the parameter values for all other attributes, namely taste, mouthfeel, aftertaste, consistency, and overall acceptability, were ranked as excellent. As the sample turmeric powder concentration increased above 1.0 g, the attributes’ rank reduced to satisfactory and fair. Thus, the fuzzy logic aided in the quantification of the subjective sensory data and facilitated the analysis and comparison of the effects of different concentrations on the attributes. No prior studies are available that have performed fuzzy logic-based sensory optimization of turmeric powder products. In a recent study, fuzzy logic analysis conveyed that the probiotic γ-aminobutyric acid has better sensory characteristics with respect to the non-probiotic bar [49]. Similar results were reported for the fuzzy logic-based sensory evaluation of carrot juice. In this case, thermosonication improved the taste and color of the carrot juice [18].

### 3.5. Nutritional Properties of the Optimal Golden Milk Formulation

As an overall conclusion of the sensory analysis, the milk system, which received the highest overall acceptability score, was chosen as the optimal product for nutritional analysis in terms of total phenolic content (TPC), total flavonoid content (TFC), antioxidant activity (AA), and curcumin content (CC). These studies were carried out for two cases, namely the unrefrigerated sample (A) and the refrigerated sample (B). Accordingly, the viability period of the golden milk product was evaluated.

Golden milk samples were prepared for these studies with 1 g of RW-dried turmeric powder mixed in 100 mL of milk. The outcome of the nutritional analysis for A and B is given in Table 3 (parts a and b). From the table, it can be seen that similar results were obtained in the case of A for RW-dried, folic acid-fortified, and NaFeEDTA-fortified turmeric powder samples. This could be because the folic acid and NaFeEDTA fortification did not adversely affect the nutritional components of the turmeric powder. However, nutritional analysis could not be carried out for the second day in case A, as the milk stored under unrefrigerated conditions, was spoiled. Similar results were obtained from a relevant study where the nutritional profile of clove-flavored milk was reported as 58.70 mg/100 g, 46.15 mg/100 g, and 487.10 mg/100 g of TPC, TFC, and AA, respectively [50].

For sample B, sample analysis indicated lower TPC, TFC, and AA values RW-dried, folic acid-fortified, and NaFeEDTA-fortified turmeric powder with an increase in the number of days (values obtained on Day 2 were lower than the values obtained on Day 1). However, the decreasing trend was not apparent for the CC as the curcumin compound was more stable than TPC, TFC, and AA compounds [7]. A reduction in TPC, TFC, and AA values was apparent for the Day 2 sample of B. However, no such analysis could be carried out on Day 3 as the golden milk sample was spoiled even after being kept in the refrigerator.

### 3.6. Storage Assessment Based on Nutritional and Physical Characteristics of Powder Products

Food products undergo dynamic alteration with respect to moisture gain or loss from the product. Such alterations exist until the system reaches a state of thermodynamic equilibrium with the surrounding environment. Thereby, the migratory role of water has a potential influence on the moisture content of food products over a period. Hence, further studies are required to analyze these aspects using storage analysis [16,51,52]. In this work, the storage studies of RW-dried, folic acid-fortified, and NaFeEDTA-fortified turmeric powder samples were conducted by storing them in a desiccator at 38 ± 1 °C for 180 days [35,51,52]. During this period, alterations in the physicochemical properties of the samples were evaluated at periodic intervals.

#### 3.6.1. Moisture Content During Storage

The MC is one of the most important factors during the evaluation of the quality and stability of RW-dried, folic acid-fortified, and NaFeEDTA-fortified turmeric powder samples. Thus, a low MC of the samples during storage affirms a decelerated and reduced rate of several degradation reactions and associated microbial growth [3]. It is well known that powder samples with high MC are susceptible to quality deterioration at higher temperatures due to the hydrolysis of oil and phospholipids, followed by an increase in sample acidity [52]. Various associated studies on the storage of powdered food products have confirmed that the optimal MC of dried powders is in the range of 4–8% for good storage stability [53,54]. During the storage study, the MC of all samples gradually increased with time, Table 4 (part a). Therefore, the powders may undergo transformations (caking) with time, and may be modified due to pertinent long-term effects of the environmental and mechanical conditions [17,55]. Since the increase in moisture content over time results from the migration of water vapor from the storage environment into the packaging material, the likelihood of caking also increases with time. However, for all tested samples, no caking was observed until 180 days of storage under ambient conditions [52].

#### 3.6.2. Antioxidant Activity During Storage

Using the DPPH method, initial AA values of the RW-dried, folic acid-fortified, and NaFeEDTA-fortified turmeric powder samples were obtained without any variation and were 90.00–89.91% (Table 4 part b). After accelerated storage, AA was reduced to about 84.80–84.00%. The AA values of RW-dried, folic acid-fortified, and NaFeEDTA-fortified turmeric powder samples did not reduce significantly with time. Hence, the pertinent losses were insignificant. The storage temperature did not alter the AA reduction trend over time. This is probably due to the AA attribute not being related to a single or several similar compounds but to a class of compounds that exhibited synergy in terms of their respective antioxidant activities [17,56]. Thus, AA showed greater interaction of various constituents, leading to the stability of AA with storage time [55].

#### 3.6.3. Total Phenolic Content During Storage

The variation in TPC with time for the tested samples is presented in Table 4c. A marginal reduction in TPC was apparent for RW-dried, folic acid-fortified, and NaFeEDTA-fortified turmeric powder samples during the storage period of 180 days. The initial total phenolic content of unfortified and fortified turmeric powders was 190.00–189.01 mg GAE/100 g. After 180 days of storage, TPC was reduced to 167.01–165.00 mg GAE/100 g. This reduction in the phenolic content was due to the oxidation of phenolic compounds along with the activation of oxidative enzymes such as polyphenoloxidase [57]. Over time, there was some increase in moisture content, and higher moisture content conveys increased molecular mobility, which may lead to higher degradation rates. Similar results have been reported for apple peel powder samples at 4, 10, and 25 °C [58], for papaya powder samples at 30 °C, and for freeze-dried strawberry puree [17,59].

#### 3.6.4. Total Flavonoid Content During Storage 

The initial TFCs of RW-dried, folic acid-fortified, and NaFeEDTA-fortified turmeric powder samples were 161.02–159.10 mg/100 g. During accelerated storage, the TFCs of the samples were reduced to 140.00–138.00 mg quercetin/100 g (Table 4 part d). Thus, the trends are like those observed for TPC. This synergy was caused by the fact that flavonoids are major phenolic compounds. Henceforth, TFC correlated with and exhibited a similar pattern to reported TPC alteration during the storage period of relevant samples [57]. Also, at room temperature, TFC may oxidize more rapidly, and this results in decreased TFC retention over storage time [60]. In the relevant literature, authors reported that the microencapsulated kenaf seed oil under accelerated storage conditions indicated a reduced TFC during storage [61]. Similarly, in another study, the authors reported a rapid degradation of TFC under high temperature conditions [57,62].

#### 3.6.5. Curcumin Content During Storage

The yellow color of turmeric is due to the polyphenolic constituent curcumin, which possesses lipophilic characteristics. In the fresh turmeric sample, the CC was 0.73% *w*/*w*. For RW-dried, folic acid-fortified, and NaFeEDTA-fortified turmeric powder samples, the curcumin content was 4.84–4.80% *w*/*w* (Table 4 part e). After the 180-days storage period, curcumin content alterations were insignificant (4.68–4.65% *w*/*w*). Another related study reported the stability of curcumin during a 28-day storage period for curcumin-fortified yogurt [63]. However, curcumin is prone to photodegradation under visible light and to oxidative degradation upon exposure to oxygen. Moreover, long-term storage leads to a gradual loss of curcumin content even in well-controlled environments [61].

#### 3.6.6. Color Indices During Storage

For RW-dried, folic acid-fortified, and NaFeEDTA-fortified turmeric powder samples, the variation of color parameters during the storage period is summarized in Table 4f. The results confirmed that the *L* value and the whiteness attribute of the tested samples remain almost unaltered during storage. However, a marginal reduction in *a* and *b* values can be observed during storage. Thus, the storage period marginally influenced the reduction of *a* and *b* values. In this regard, it is known that while *b* values indicate yellowness (+) or blueness (−), *a* values represent the redness (+) or greenness (−) of the tested samples. The color of food products during storage is affected by various factors, including packaging material, storage temperature, sugar and protein composition, water activity, and storage time. Also, curcumin content critically contributes to the turmeric color due to the pigmented curcumin compounds that impart its characteristic yellow color. Therefore, good curcumin stability in dried turmeric led to good color stability of the product [63].

#### 3.6.7. Folic Acid Content During Storage

The folic acid content variation in folic acid-fortified turmeric samples during accelerated storage conditions is summarized in Table 4 part g. For a storage time variation of up to 180 days, the total folic acid content reduced from 20.00 to 18.74 mg/100 g of the sample of turmeric powder. Such a marginal reduction in folic acid content was due to the stable nature of the folic acid used for fortification [64]. Another study reported similar results for the stability of folic acid after fortification in blackberry powder [65]. Also, the physicochemical characteristics of the RW-dried turmeric powder did not alter significantly due to folic acid fortification.

#### 3.6.8. Iron Content During Storage

For the NaFeEDTA-fortified RW-dried turmeric sample, the variation in iron content during storage conditions is summarized in Table 4 part h. Over time, an insignificant reduction in iron content was apparent. The iron content ranged from 20.00 to 19.41 mg/100 g of the sample. Such an insignificant reduction is due to the stability of the turmeric powder and NaFeEDTA powder. This aspect did not result in significant interaction. Also, the inorganic and stable nature of iron powder fostered a minimum loss of iron content over time in the NaFeEDTA fortified turmeric sample [66].

#### 3.6.9. Bulk Density During Storage

The bulk density of the turmeric powder samples increased with storage time (Table 4 part i). This was due to the moisture gain of the powder samples. A similar enhancement in the bulk densities with increasing MC during storage has been reported in previous studies for the mango milk powder system [67]. An increase in bulk density may also be attributed to enhanced cohesiveness between powder particles due to enhanced absorption of moisture during the storage period. Also, prolonged storage leads to particle settling and compaction, which in turn increases bulk density. However, the bulk density was stable for folic acid-fortified spray-dried blackberry in a related prior study [65].

#### 3.6.10. Solubility During Storage

The alterations in solubility of RW-dried, folic acid-fortified, and NaFeEDTA-fortified turmeric powder samples during storage at various RH levels are presented in Table 4 part j. After RW drying, the turmeric powders possessed a solubility of 30–29.00 % and underwent almost complete dissolution in water. However, with time, the solubility of all tested samples did not change up to the 9th week. Thereafter, the parameter reduced marginally to 25–26.00 % at the end of the storage period (week 24). These findings confirmed that the solubility of all samples was influenced by the relative humidity conditions of the storage environment, and a higher RH level contributed to marginal solubility loss [3]. With an increase in storage time, there was a marginal enhancement in the moisture content of the sample. This eventually led to a loss in solubility. This insignificant solubility loss was due to a minor enhancement in moisture content that has an adverse influence on the solubility. Another possible reason could be the protein–carbohydrate interaction (Maillard browning), which reduces solubility over time [68].

#### 3.6.11. Swelling Power During Storage

The swelling power of RW-dried, folic acid-fortified, and NaFeEDTA-fortified turmeric powder samples varied marginally during the duration of the storage period (Table 4 part k). During the initial storage period, the swelling power remained constant at 1.8 g/g (up to week 9). However, after the 9th week, a marginal reduction in swelling power was noticed. This parametric reduction did not significantly increase with time during the later stages of storage. Thus, for a maximum storage period of 180 days, a reduction in swelling power was not significant (value of 1.5 g/g). The lower reduction in swelling power over the storage period was due to the stable nature of the turmeric powder. Since all tested samples did not significantly absorb moisture content, their stability remained intact in terms of good swelling power during the total time period of the storage study [68].

#### 3.6.12. Water Binding Capacity During Storage

The water binding capacity of RW-dried, folic acid-fortified, and NaFeEDTA-fortified turmeric powder samples is summarized in Table 4 part l. Long-term storage leads to gradual structural changes (e.g., crystallization of amorphous sugars), which reduces the WHC. The table infers that with storage time, a marginal reduction in water binding capacity occurred. However, the reduction was insignificant. Up to week 9, water binding capacity remained unchanged (66%). However, after week 17, a reduction was apparent but marginal. This may be due to the stable nature of the tested samples that absorbed less moisture during storage. Hence, high water binding capacity could be retained even at longer storage period conditions [23].

#### 3.6.13. Dispersion Time During Storage

The reconstitution properties of powdered food products ensured their acceptability to the consumer market. Dispersion is one of the most important reconstitution properties. It is defined as the pace at which the powder dissolves upon reconstitution in water. In other words, any powder material can be inferred to be of the best quality for the scenario in which the sample’s particles are immediately dissolved in water without the creation of lumps. The dispersion time of the RW-dried, folic acid-fortified, and NaFeEDTA-fortified turmeric powder samples varied between 20 and 17 s (Table 4 part m). The reduced dispersibility over longer storage periods could be due to the powder’s rising moisture content, which possibly enhances particle cohesiveness and henceforth facilitates lump formation during water dissolution [44]. Also, insignificant differences have been recorded in the dispersibility of RW-dried, folic acid-fortified, and NaFeEDTA-fortified turmeric powder samples after the second week. Therefore, good confidence levels have been ensured, and the findings corroborate relevant prior work [23,52].

#### 3.6.14. Hygroscopicity During Storage

Table 4 part n summarizes the findings associated with the hygroscopicity of RW-dried, folic acid-fortified, and NaFeEDTA-fortified turmeric powder samples subjected to accelerated storage conditions for up to 180 days. The initial hygroscopicity of RW-dried, folic acid-fortified, and NaFeEDTA-fortified turmeric powder samples was 8.80–8.70 g/100 g. Thereafter, the parameter value was reduced to 8.10–8.00 g/100 g during accelerated storage conditions. While the powder initially exhibited minimal moisture uptake, prolonged storage resulted in progressive moisture absorption [15]. This was probably due to the enhanced moisture content in the powder that eventually led to further reduction in the absorbed water [68].

## 4. Conclusions

The present work provides a benchmark for the utilization of RW-dried turmeric, folic acid, and iron in golden milk formulations. Also, the study demonstrates an in-depth shelf-life analysis of golden milk formulations and fortified powder mix with various quality parameters. The sorption characteristics confirmed that the GAB model has the best fit for fortified and unfortified turmeric samples. The fuzzy logic-based sensory analysis indicates that the formulation with 1 g of unfortified turmeric powder in 100 mL of milk is the best formulation. The various quality fractions assessed during the storage study were not significantly altered for the RW-dried, folic acid-fortified, and NaFeEDTA-fortified turmeric powder. This indicates the superior shelf life and applicability of the prepared powder mixes. The results indicated that for the fortified turmeric powder mix, the reduction of quality parameters was marginal and was not significant for a duration of up to 24 weeks. For the case of the golden milk formulations up to 48 h, there was no significant loss of nutritional parameters. In the future, researchers may explore the fortification of folic acid and NaFeEDTA in similar staple foods with detailed shelf-life studies. This will pave the way for newer horizons in functional food products.

## Figures and Tables

**Figure 1 foods-14-02948-f001:**
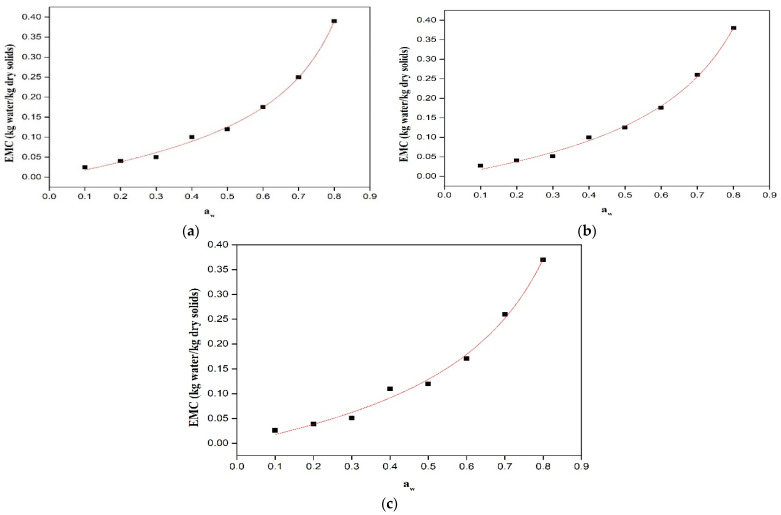
Moisture content alteration and water activity for (**a**) refractance window (RW)-dried, (**b**) folic acid-fortified RW-dried, and (**c**) NaFeEDTA-fortified RW-dried turmeric powder samples.

**Figure 2 foods-14-02948-f002:**
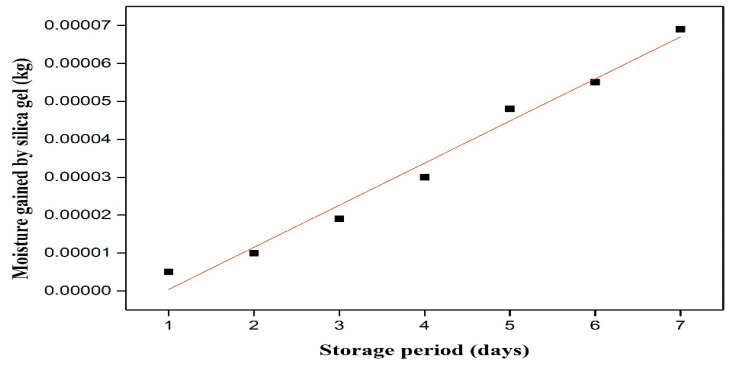
Permeability plot for moisture gains by silica gel over time.

**Figure 3 foods-14-02948-f003:**
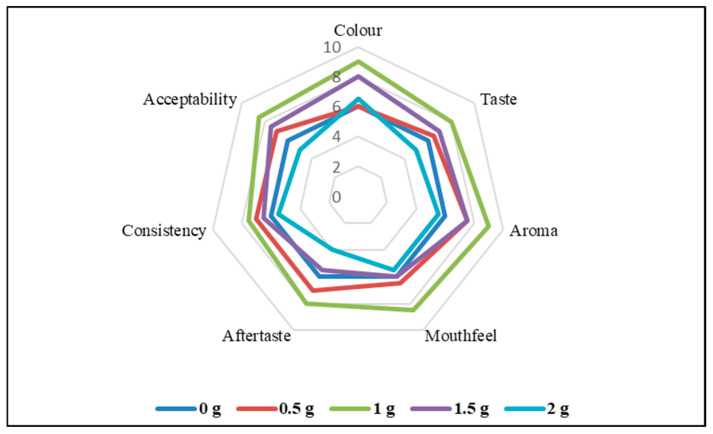
Radar chart depicting sensory characteristics of refractance window-dried turmeric powder-based milk products for varied turmeric constitution cases.

**Table 1 foods-14-02948-t001:** GAB model fitness parameters for refractance window (RW)-dried, folic acid-fortified RW-dried, and NaFeEDTA-fortified RW-dried turmeric powder samples.

S. No.	Samples	C	K	Mo	R^2^
1.	Unfortified	0.476	0.996	0.358	0.99
2.	Folic acid fortified	0.650	0.920	0.271	0.99
3.	NaFeEDTA fortified	0.630	0.931	0.280	0.98

**Table 2 foods-14-02948-t002:** Similarity values ranking of sensory attributes based on fuzzy logic.

Sl. No.	Attribute	0 g	0.5 g	1 g	1.5 g	2 g
1.	Color	0.1757	0.6377	0.8038	0.8250	0.6769
2.	Taste	0.2233	0.7648	0.8813	0.7863	0.5181
3.	Aroma	0.2043	0.7017	0.8436	0.8255	0.6426
4.	Mouthfeel	0.2239	0.7669	0.8832	0.7865	0.5164
5.	Aftertaste	0.2903	0.8732	0.9283	0.7351	0.4115
6.	Consistency	0.2239	0.7669	0.8832	0.7865	0.5164
7.	Overall Acceptability	0.2160	0.7394	0.8690	0.8114	0.5816

**Table 3 foods-14-02948-t003:** A summary of nutritional characteristics of (**a**) unrefrigerated golden milk products and (**b**) refrigerated golden milk products.

**(a)**						
**S. No.**	**Day**	**Samples**	**AA (%)**	**TPC (mg GAE/100 mL)**	**TFC (mg Quercetin/100 mL)**	**CC (% *w*/*w*)**
1.	0	Unfortified	24.50 ± 2	876.21 ± 1	784.61 ± 2	4.20 ± 0.1
2.		Folic acid fortified	23.50 ± 1	878.01 ± 3	788.97 ± 1	4.21 ± 0.1
3.		NaFeEDTA fortified	24.10 ± 2	870.87 ± 1	787.49 ± 2	4.19 ± 0.2
**(b)**						
**S. No**	**Day**	**Samples**	**AA** **(%)**	**TPC (mg GAE/100 mL)**	**TFC (mg Quercetin/100 mL)**	**CC (% *w*/*w*)**
1.	1	Unfortified	22.50 ± 1	830.14 ± 3	710.76 ± 2	4.14 ± 0.1
2.		Folic acid fortified	22.80 ± 2	825.51 ± 3	716.87 ± 3	4.16 ± 0.2
3.		NaFeEDTA fortified	22.68 ± 1	829.84 ± 3	717.54 ± 3	4.14 ± 0.1
1.	2	Unfortified	20.90 ± 2	784.29 ± 3	650.71 ± 3	4.10 ± 0.1
2.		Folic acid fortified	20.12 ± 2	786.31 ± 2	646.15 ± 3	4.10 ± 0.2
3.		NaFeEDTA fortified	21.00 ± 2	781.62 ± 2	649.25 ± 2	4.11 ± 0.1

**Table 4 foods-14-02948-t004:** Time-dependent data of stored refractance window-dried turmeric powder products.

**(a) Moisture Content (%)**													
**S. No.**	**Sample**	**0 Weeks**			**9 Weeks**			**17 Weeks**			**24 Weeks**		
1.	Unfortified	4.00 ± 0.1			4.90 ± 0.1			6.10 ± 0.1			7.81 ± 0.2		
2.	Folic acid fortified	4.00 ± 0.2			4.95 ± 0.2			6.10 ± 0.1			7.80 ± 0.2		
3.	NaFeEDTA fortified	4.00 ± 0.3			4.91 ± 0.2			6.30 ± 0.1			7.83 ± 0.3		
**(b) Anti-Oxidant Activity (%)**													
**S. No.**	**Sample**	**0 Weeks**			**9 Weeks**			**17 Weeks**			**24 Weeks**		
1.	Unfortified	90.00 ± 0.5			88.60 ± 0.6			85.40 ± 0.3			84.80 ± 0.3		
2.	Folic acid fortified	89.91 ± 0.6			88.70 ± 0.2			85.90 ± 0.5			84.00 ± 0.4		
3.	NaFeEDTA fortified	89.97 ± 0.6			87.40 ± 0.5			85.10 ± 0.2			84.50 ± 0.5		
**(c) Total Phenolic Content (mg GAE/g sample)**													
**S. No.**	**Sample**	**0 Weeks**			**9 Weeks**			**17 Weeks**			**24 Weeks**		
1.	Unfortified	189.76 ± 2			180.00 ± 3			175.23 ± 2			167.01 ± 2		
2.	Folic acid fortified	190.00 ± 2			179.12 ± 2			176.31 ± 2			165.00 ± 2		
3.	NaFeEDTA fortified	189.01 ± 4			181.15 ± 2			173.47 ± 2			166.25 ± 2		
**(d) Total Flavonoid Content (mg quercetin/g sample)**													
**S. No.**	**Sample**	**0 Weeks**			**9 Weeks**			**17 Weeks**			**24 Weeks**		
1.	Unfortified	160.00 ± 4			153.00 ± 5			145.00 ± 2			139.00 ± 4		
2.	Folic acid fortified	159.10 ± 3			152.00 ± 5			147.00 ± 2			138.00 ± 3		
3.	NaFeEDTA fortified	161.02 ± 2			151.00 ± 4			146.00 ± 1			140.00 ± 3		
**(e) Curcumin Content (% *w*/*w*)**													
**S. No.**	**Sample**	**0 Weeks**			**9 Weeks**			**17 Weeks**			**24 Weeks**		
1.	Unfortified	4.84 ± 0.02			4.75 ± 0.03			4.70 ± 0.02			4.65 ± 0.01		
2.	Folic acid fortified	4.80 ± 0.01			4.76 ± 0.03			4.71 ± 0.02			4.67 ± 0.01		
3.	NaFeEDTA fortified	4.83 ± 0.02			4.77 ± 0.02			4.72 ± 0.02			4.68 ± 0.02		
**(f) Color Indices**													
**S. No.**	**Sample**	**0 Weeks**			**9 Weeks**			**17 Weeks**			**24 Weeks**		
		**L**	**a**	**b**	**L**	**A**	**b**	**L**	**a**	**b**	**L**	**a**	**b**
1.	Unfortified	56 ± 2	31 ± 1	62 ± 3	55 ± 2	29 ± 1	60 ± 2	53 ± 1	26 ± 3	57 ± 2	51 ± 2	23 ± 1	53 ± 3
2.	Folic acid fortified	55 ± 2	30 ± 2	61 ± 3	54 ± 2	29 ± 1	60 ± 2	52 ± 3	26 ± 3	56 ± 1	51 ± 2	23 ± 1	53 ± 2
3.	NaFeEDTA fortified	56 ± 2	30 ± 1	61 ± 1	54 ± 1	29 ± 1	60 ± 1	53 ± 1	26 ± 3	57 ± 1	52 ± 2	23 ± 1	53 ± 2
**(g) Folic Acid (mg/100 g sample)**													
**S. No.**	**Sample**	**0 Weeks**			**9 Weeks**			**17 Weeks**			**24 Weeks**		
1.	Folic acid fortified	20.00 ± 0.5			19.65 ± 0.5			19.01 ± 1			18.74 ± 1		
**(h) Iron Content (mg/100 g sample)**													
**S. No.**	**Sample**	**0 Weeks**			**9 Weeks**			**17 Weeks**			**24 Weeks**		
1.	NaFeEDTA fortified	20.00 ± 0.1			19.91 ± 0.2			19.72 ± 0.2			19.41 ± 0.3		
**(i) Bulk Density (g/mL)**													
**S. No.**	**Sample**	**0 Weeks**			**9 Weeks**			**17 Weeks**			**24 Weeks**		
1.	Unfortified	0.62 ± 0.01			0.64 ± 0.04			0.67 ± 0.04			0.69 ± 0.01		
2.	Folic acid fortified	0.65 ± 0.01			0.66 ± 0.05			0.68 ± 0.02			0.70 ± 0.03		
3.	NaFeEDTA fortified	0.64 ± 0.03			0.65 ± 0.05			0.67 ± 0.03			0.69 ± 0.03		
**(j) Solubility (%)**													
**S. No.**	**Sample**	**0 Weeks**			**9 Weeks**			**17 Weeks**			**24 Weeks**		
1.	Unfortified	29.00 ± 1			29.00 ± 2			27.00 ± 1			26.00 ± 3		
2.	Folic acid fortified	30.00 ± 3			30.00 ± 2			28.00 ± 1			27.00 ± 3		
3.	NaFeEDTA fortified	28.00 ± 1			28.00 ± 3			26.00 ± 1			25.00 ± 2		
**(k) Swelling Power (g/g)**													
**S. No.**	**Sample**	**0 Weeks**			**9 Weeks**			**17 Weeks**			**24 Weeks**		
1.	Unfortified	1.80 ± 0.1			1.80 ± 0.2			1.70 ± 0.3			1.50 ± 0.1		
2.	Folic acid fortified	2.00 ± 0.1			2.00 ± 0.2			1.90 ± 0.2			1.70 ± 0.3		
3.	NaFeEDTA fortified	1.90 ± 0.2			1.90 ± 0.2			1.80 ± 0.3			1.60 ± 0.3		
**(l) Water Holding Capacity (%)**													
**S. No.**	**Sample**	**0 Weeks**			**9 Weeks**			**17 Weeks**			**24 Weeks**		
1.	Unfortified	66.00 ± 1			66.00 ± 2			64.00 ± 1			61.00 ± 2		
2.	Folic acid fortified	65.00 ± 2			65.00 ± 2			63.00 ± 1			59.00 ± 1		
3.	NaFeEDTA fortified	67.00 ± 2			67.00 ± 2			65.00 ± 1			60.00 ± 1		
**(m) Dispersion Time (s)**													
**S. No.**	**Sample**	**0 Weeks**			**9 Weeks**			**17 Weeks**			**24 Weeks**		
1.	Unfortified	20 ± 2			22 ± 2			26 ± 1			30 ± 2		
2.	Folic acid fortified	17 ± 2			21 ± 2			25 ± 1			29 ± 2		
3.	NaFeEDTA fortified	19 ± 2			23 ± 2			28 ± 1			31 ± 2		
**(n) Hygroscopicity**													
**S. No.**	**Sample**	**0 Weeks**			**9 Weeks**			**17 Weeks**			**24 Weeks**		
1.	Unfortified	8.70 ± 0.1			8.60 ± 0.1			8.40 ± 0.2			8.10 ± 0.1		
2.	Folic acid fortified	8.80 ± 0.2			8.60 ± 0.1			8.50 ± 0.2			8.00 ± 0.2		
3.	NaFeEDTA fortified	8.50 ± 0.2			8.40 ± 0.2			8.30 ± 0.1			8.00 ± 0.2		

## Data Availability

The original contributions presented in the study are included in the article, further inquiries can be directed to the corresponding author.

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
