# Peer review of "Development and Fuzzy Logic-Based Optimization of Golden Milk Formulations Using RW-Dried Turmeric Powder: A Study on Shelf Life, Sensory Attributes, and Functional Properties"

_foods, 2025, doi:10.3390/foods14172948_

Round 1

Reviewer 1 Report

Comments and Suggestions for Authors

The research paper entitled "Development and Fuzzy Logic based Optimization of Golden 1 Milk Formulations using RW-Dried Turmeric Powder: A Study 2 on Shelf Life, Sensory Attributes, and Functional Properties" assess the shelf life of refractance window dried turmeric powder with and without NaFeEDTA.

General Comments 

The research paper presents an interesting approach to the storage of food storage conditions. Howerver, in my opinion, the motivation for the study should be better described in the abstract. Furthermore, the english writing must be revised. The authors should also provide their critical opinion and better discuss their results on each section instead of only mentioning the obtained data. 

Topic 2.2.: A better description of the method should be provided. Were the optimization of the method done with Tumeric? If so, what does this study addresses that is not addressed in the optimization study? If this research paper is a continuation of the work of Daulay et al., 2019, then specific information should be provided to differentiate both studies.  

Topic 2.8 should provide a detailed description of the characterization methods.

A topic refering the statiscal analysis should be provided.  

In my opinion, in the 3.4, the results regarding the sensory tests should be presented as a function of concentration, i.e., one chart for each sensory profile or a spider chart and a discussion should be made regarding the full profile of the product and not each individual parameter. 

Specific Comments

Line 21: The term "total phenolic content (TPC)" is duplicate 

Line 35: subtitute are with is (refers to the understanding)

Line 36: Suggestion: "Studying food storage" or "Food Storare assays" insteas of "Storage study"

Line 38: The same as above

Line 40: Substitute "the duration of period" with "the amount of time"

line 83: the authors forgot to include the examples of food powders in the sentence: "the shelf life of various food powders like"

line 137: The reference should be right next to the text that mentions that the optimal conditions were determined elsewhere. 

line 230 - 231: The text is deformated

In topic 3.2 a critical discussion of the resutls should be provided. What is the meaning of the permeability number? Does it indicate potential applications for the package material? Can this topic be included on another discussion? 

Line 430 and line 449: revise this line

Author Response

Response to reviewer’s comment

We thank the reviewers for their valuable comments and useful insights that enabled the manuscript to undergo a major revision. A detailed account of the responses for all comments is as follows:

Reviewer #1

The research paper entitled "Development and Fuzzy Logic based Optimization of Golden 1 Milk Formulations using RW-Dried Turmeric Powder: A Study 2 on Shelf Life, Sensory Attributes, and Functional Properties" assess the shelf life of refractance window dried turmeric powder with and without NaFeEDTA.

General Comments 

Comment 1: The research paper presents an interesting approach to the storage of food storage conditions. However, in my opinion, the motivation for the study should be better described in the abstract. Furthermore, the English writing must be revised. The authors should also provide their critical opinion and better discuss their results on each section instead of only mentioning the obtained data. 

Response: We thank the reviewer for the comment and changes have been made accordingly in the revised manuscript.

Line 14 - 36

The storage characteristics of folic acid and NaFeEDTA fortified in a refractance window dried turmeric powder base and its subsequent application in the formulation of nutritionally functionalized golden milk has not been addressed in previous studies. The golden milk is a staple food and ideal matrix for the fortification of important nutrients such as iron and folic acid. With this motivation, the present study assesses the refractance window (RW) dried turmeric powder fortified with folic acid and NaFeEDTA in terms of the moisture isotherm, permeability of packing material, and storage parameters to calculate the shelf life. Further, a sensory analysis was conducted based on the Fuzzy logic method to obtain the best constitution of RW dried turmeric powder in milk. Thereby, for the best formulation of golden milk, characteristics of the product under unrefrigerated and refrigerated conditions have been evaluated in addition to the storage study. Moreover, moisture content (MC), total flavonoid content (TFC), total phenolic content (TPC) antioxidant activity (AA), curcumin content (CC), colour indices, bulk densities, water binding, and capacities have been studied with respect to the time alteration. The results inferred upon a healthy shelf life of 184, 187, and 183 days for RW dried turmeric powder, folic acid fortified, and NaFeEDTA fortified RW dried turmeric powder samples, respectively, in the zipper pouch system.The fuzzy scores ranked the sample with 1g concentration of turmeric powder as the best, considering taste, aroma, mouthfeel, aftertaste, consistency, and overall acceptability. The TPC, TFC, AA, and CC values for RW dried turmeric powder in milk were 876.21 mg GAE/100 mL, 784.61 mg quercetin/100 mL, 24.50 % and 4.20 % w/w, respectively. Marginal alterations have been affirmed for the RW-dried fortified and unfortified turmeric samples. This fortified golden milk has the potential to be used as a health drink.

Comment 2: Topic 2.2.: A better description of the method should be provided. Were the optimization of the method done with Turmeric? If so, what does this study addresses that is not addressed in the optimization study? If this research paper is a continuation of the work of Daulay et al., 2019, then specific information should be provided to differentiate both studies.  

Response: We thank the reviewer for the comment and changes have been made accordingly in the revised manuscript. This research is not a continuation of Daulay et al., 2019. It is a continuation of our earlier work which was published in MDPI-Foods titled “Talukdar, P., Baruah, K. N., Barman, P. J., Sharma, S., & Uppaluri, R. V. (2025). Development and Characterization of Refractance Window-Dried Curcuma longa Powder Fortified with NaFeEDTA and Folic Acid: A Study on Thermal, Morphological, and In Vitro Bio Accessibility Properties. Foods14(4), 658.”

Line 146 – 162

The refractance window drying (RWD) experiments were conducted using response surface methodology (RSM) for the 1 mm turmeric slice sample. The experimental design was based on the Box Behnken Design (BBD) methodology being applied using Design expert software (Version 7). The BBD based RSM design indicated 15 experimental data sets in which 3 sets correspond to those being investigated at the central point. The BBD enabled better precision in the centered factor space. Thus, 3-factors-3-level BBD with three runs at the center point was adopted for RWD investigation of the turmeric slice samples. Based on initial investigations, the range of water bath temperature, and air-velocity and drying time range have been considered as 65 – 95°C, 0.5 – 1 m/s and 75 – 360 min respectively. After the conduct of the RSM experiments, the optimized water bath temperature, and air-velocity and drying time have been set as 95 °C, 0.75 m/s, and 75 min, respectively [7]. Thereafter, the dried samples were powdered using a dry portable electric grinder. Eventually, sieved samples were obtained through the 80-mesh sieve and were analyzed to possess an average particle size of 0.177 mm [28]. Also, it shall be noted that the optimization studies were conducted with the turmeric slices and fortification was not carried out during such conduct of the optimization studies.  Thus, the conduct of the investigations was followed as a follow up work of our earlier article [6].

Comment 3: Topic 2.8 should provide a detailed description of the characterization methods. A topic referring the statistical analysis should be provided.  

Response: We thank the reviewer for the comment and changes have been made accordingly in the revised manuscript.

Line 247 – 296

2.8. Characterization of Optimal Gold Milk Formulation

For the sensory analysis-based optimal gold milk formulation, relevant characterizations were carried out in terms of AA, TPC, TFC, and CC. The adopted procedures have been briefly summarized in the following sub-sections.

 2.8.1 Total Phenolic Content

The total phenolic content was estimated with the FCR method [35]. The sample processing procedure involved the mixing 1 mL of golden milk with 1 mL of prepared FCR reagent in a vortex mixer. After 5 minutes of mixing, 10 mL of 7 % sodium carbonate solution was added to the mixture. After subsequent vortex mixing, the final solution volume was adjusted to 25 mL. Finally, the solution was kept in a dark environment for 2 h at ambient temperature. This was followed by the absorbance determination for the sample at 750 nm and with the UV spectrophotometer (UV-2600, Shimadzu, Singapore). Using a calibration chart being prepared with gallic acid, the TPC content of the sample was expressed in terms of mg of GAE/g sample.

2.8.2 Total Flavonoid Content

AlCl3 method was duly followed to determine the TFC [35]. Firstly, this involved vortex mixing of 1 mL of golden milk with 4 mL distilled water. Thereafter, 0.3 mL of 5 % NaNO3 solution was added and the mixture was again subjected to vortex mixing for 5 minutes. Eventually, 0.3 mL of 10 % AlCl3 was added to the mixture for subsequent vortex mixing for 6 minutes. Later, 2 mL of 1M NaOH was added to the solution and the mixture was subjected to thorough mixing.  The final solution with a volume of 10 mL was kept in a dark environment for 30 minutes at ambient temperature. Thereafter, the mixture absorbance was measured at 517 nm. Using quercetin based calibration chart, the TFC of the sample was determined in terms of mg of quercetin equivalent/g sample.

2.8.3 Anti-oxidant Activity

The golden milk product was subjected to total antioxidant activity evaluation and with the 1,1-diphenyl-2-picryl hydrazyl (DPPH) assay method  [35]. To do so, firstly, 2 mL golden milk was mixed with 2 mL DPPH stock solution (1:1). Simultaneously, a control sample was prepared using distilled water and DPPH stock solution. Thereafter, the samples were incubated at 30 min in a dark environment at room temperature. Eventually, the samples were analysed for their absorbance at 510 nm. Thereby, the % AA was determined using the expression:

where Ac and As corresponds to the absorbance of control and sample respectively.

2.8.4 Curcumin Content

The curcumin content of the golden milk was determined by adopting the procedure summarized in [35]. Firstly, 1 mL of milk was placed in a test tube and was boiled with 10 mg dried turmeric sample. Thereafter, the sample was cooled to room temperature. Thereafter, to the sample, 5 mL of 95 % methanol was added. After subsequent sonication for 10 min, 5 mL of extract was mixed with 95 % methanol to achieve a solution volume of 10 mL. Thereafter, the solution was filtered using Whatman filter paper No. 1. After filtration, 0.4 mL of filtrate was added to 5 mL of 95 % methanol. The absorbance of the samples was measured at 425 nm and with the UV-VIS spectrophotometer. Using the measured absorbance of the standard and turmeric processed samples, the curcumin content of the golden milk was determined with the calibration curve (in terms of % curcumin content w/w).

2.9. Experimental design and statistical analysis

            The experiments followed a completely randomized design with three replicates. The obtained data was analyzed with the analysis of variance (ANOVA) and for a significance level of p as 0.05 with the Origin Pro 9 software. [6,7]

Comment 4: In my opinion, in the 3.4, the results regarding the sensory tests should be presented as a function of concentration, i.e., one chart for each sensory profile or a spider chart and a discussion should be made regarding the full profile of the product and not each individual parameter. 

Response: We thank the reviewer for the comment and changes have been made accordingly in the revised manuscript.

Line 379 – 419

3.4. Sensory Assessment of Powder Product

3.4.1 Hedonic scale based sensory analysis

As a simple marketing rule, if consumers do not like the appearance, flavour, or texture of a food product, they will not buy it. Therefore, the overall sensory experience of a product is crucial for its commercial success. Specific protocols and methods have been developed to estimate and quantify consumers' sensory experiences. Accordingly, the risk associated with the non-acceptability of a food product can be reduced through the scientific relevance in terms of the sensory descriptive analysis or sensory descriptive evaluation. Appropriate sensory evaluation allows a very useful understanding of the key attributes that assist in the commercial success of food products [47].

The RW dried turmeric powder has been evaluated for its sensory characteristics. However, the folic acid fortified and NaFeEDTA fortified turmeric powder samples were not analysed for their sensory characteristics. This is due to the non-food grade status of the deployed folic acid and NaFeEDTA fortificants. The sensory analysis was conducted with a panel of judges and in terms of the scores provided for colour and appearance, taste, aroma, mouthfeel, aftertaste, consistency, and overall acceptability [48].

The colour and appearance of a product are a prime entity for its formal acceptance or rejection during sensory analysis. The aesthetic quality of a food product is dominantly influenced by its colour and thereby provides visual inputs for flavour identification. After colour and appearance, aroma has significance in the sensory evaluation of a sample and thereby critically influences the formal acceptance or rejection of a sample by a consumer. The smell or aroma of a product influences the olfactory glands of consumers and accordingly enhances the desire of the consumers to taste a product. Thus, a product with a good aroma attracts a consumer to taste it, and a bad or strong aroma hinders the formal acceptance of the food product.  Despite having good appearance and colour properties, a food product is usually accepted for its good taste. The aftertaste is an attribute that lingers in the palate of the mouth after a sample's ingestion into the mouth. The aftertaste of the product is an important sensory attribute and reflects upon the acceptability of the product by the consumers. The consistency and mouthfeel of a product is an important sensory attribute to infer upon its degree of acceptability and product's acceptability. However, the overall acceptability parameter has the final say in the acceptability of a product by the consumer[47,48].

The sensory analysis was performed for 100 ml milk mixed with 0g, 0.5, 1g, 1.5g and 2 g of RW dried turmeric powder. Twenty-five semi trained panellists had evaluated the samples for the sensory attributes - colour and appearance, aroma, taste, aftertaste, consistency, mouthfeel and overall acceptability – and with the 9 score hedonic scale (Fig 3). At lower concentrations, i.e., 0 g and 0.5 g, the sensory scores for colour were liked marginally. However, other attributes were moderately liked. The sensory scores for 1 g concentration were highest between 8 to 9 (Like very much and like moderately) for all the attributes. At 2g, the scores for mouthfeel, aftertaste, and overall acceptability suggested that the sample is disliked by panellists.

 Figure 3. Radar chart depicting sensory characteristics of refractance window dried turmeric powder-based milk products for varied turmeric constitution cases

Specific Comments

Comment 5: Line 21: The term "total phenolic content (TPC)" is duplicate 

Response: We thank the reviewer for the comment and changes have been made accordingly in the revised manuscript.

Line 25 – 27

Moreover, moisture content (MC), total flavonoid content (TFC), total phenolic content (TPC) antioxidant activity (AA), curcumin content (CC), colour indices, bulk densities, water binding capacities have been studied with respect to the time alteration.

Comment 6: Line 35: subtitute are with is (refers to the understanding)

Response: We thank the reviewer for the comment and changes have been made accordingly in the revised manuscript.

Line 41 - 42

The understanding of storage stability and shelf life of food products is essential to ensure the product quality, safety, and efficacy of the assessed sample with time.

Comment 7: Line 36: Suggestion: "Studying food storage" or "Food Storare assays"

insteas of "Storage study"

Response: We thank the reviewer for the comment and changes have been made accordingly in the revised manuscript.

Line 42 – 44

Food storage study is important as it helps in evaluating the effect of environmental factors like temperature, light, and humidity on the food products [1].

Comment 8: Line 38: The same as above

Response: We thank the reviewer for the comment and changes have been made accordingly in the revised manuscript.

Line 44 – 46

Also, from the food storage assays, the influence of packaging on microbiological, chemical, and physical attributes of food products can be determined.

Comment 9: Line 40: Substitute "the duration of period" with "the amount of time"

Response: We thank the reviewer for the comment and changes have been made accordingly in the revised manuscript.

Line 46 – 47

The determination of shelf life helps to estimate the amount of time duration for which the food product remains viable for consumption.

Comment 10: line 83: the authors forgot to include the examples of food powders in the sentence: "the shelf life of various food powders like"

Response: We thank the reviewer for the comment and changes have been made accordingly in the revised manuscript.

Line 93 – 95

Food powder mixes are usually sensitive to storage environment factors such as moisture, light exposure, oxygen, and heat[15], [16]. These environmental variations potentially affect the shelf life of various food powders [17] [18].

Comment 11: line 137: The reference should be right next to the text that mentions that the optimal conditions were determined elsewhere. 

Response: We thank the reviewer for the comment and changes have been made accordingly in the revised manuscript.

Line 146 – 159

The refractance window drying (RWD) experiments were conducted using response surface methodology (RSM) for the 1 mm turmeric slice sample. The experimental design was based on the Box Behnken Design (BBD) methodology being applied using Design expert software (Version 7). The BBD based RSM design indicated 15 experimental data sets in which 3 sets correspond to those being investigated at the central point. The BBD enabled better precision in the centered factor space. Thus, 3-factors-3-level BBD with three runs at the center point was adopted for RWD investigation of the turmeric slice samples. Based on initial investigations, the range of water bath temperature, and air-velocity and drying time range have been considered as 65 – 95°C, 0.5 – 1 m/s and 75 – 360 min respectively. After the conduct of the RSM experiments, the optimized water bath temperature, and air-velocity and drying time have been set as 95 °C, 0.75 m/s, and 75 min, respectively [7]. Thereafter, the dried samples were powdered using a dry portable electric grinder. Eventually, sieved samples were obtained through the 80-mesh sieve and were analyzed to possess an average particle size of 0.177 mm [28].

Comment 12: line 230 - 231: The text is deformated

Response: We thank the reviewer for the comment and changes have been made accordingly in the revised manuscript.

Line 235 – 246

Fuzzy logic was performed to analyze the sensory scores of the samples. The first step involved is the calculation of the triplet for all the sensory scores for each attribute (using equation 4). The relative weightages of quality attributes were calculated (using equation 5), and the product of two triplets, say example, T1 and T2, were determined with the expression presented in equation 6. The overall scores, Oattr for an attribute is the sum of the products of its triplet with the relative weightages of all attributes (as shown in equation 7). The membership value µ for a triplet , and a score x was calculated with the following expression (equation 8). The membership on a standard Fuzzy scale was determined for each x in the range [0, 100] and with steps for n number of samples (using equation 9). The similarity S between the sample values and a standard fuzzy scale was calculated with the following expression (Equation 10). Further, the rankings were based on the maximum similarity value for each attribute (using equation 11).

Comment 13: In topic 3.2 a critical discussion of the resutls should be provided. What is the meaning of the permeability number? Does it indicate potential applications for the package material? Can this topic be included on another discussion? 

Response: We thank the reviewer for the comment and changes have been made accordingly in the revised manuscript.

Line 344 – 351

The preliminary study was carried out to find the suitability of the packaging material (Low density polyethylene i.e zipper pouch) for packaging of turmeric powder. Fig. 2 depicts the time-dependent cumulative moisture gain with time by silica gel in a zipper pouch at 38 ± 2 oC and 90 % RH. Thereby, the slope of the straight line fit (dw/dqp) was 0.000011 kg/day. Using expression for permeability determination, the water vapour permeability (K) of the zipper pouch (Low density polyethylene) for a surface area value of 0.014 m2 and p* value of 6980.5 Pa (saturated vapor pressure of water at 38 oC), was obtained as 1.12 ×10-7 kg/m2dayPa [10].

Comment 14: Line 430 and line 449: revise this line

Response: We thank the reviewer for the comment and changes have been made accordingly in the revised manuscript.

Line 444 – 473

3.5. Nutritional Properties of the Optimal Golden Milk Formulation

As an overall conclusion of the sensory analysis, the milk system that received the highest overall acceptability score was chosen as the optimal product for nutritional analysis in terms of total phenolic content (TPC), total flavonoids content (TFC), antioxidant activity (AA), and curcumin content (CC). These studies were carried out for two cases, namely the unrefrigerated sample (A) and the refrigerated sample (B). Accordingly, the viability period of the golden milk product were evaluated.

Golden milk samples were prepared for these studies. To do so, 1 g of RW dried turmeric powder was mixed in 100 mL of milk. The outcome of the nutritional analysis for A and B are given in Table 3 (a) and (b). From the table, it is seen that similar results were obtained in case of A for RW dried turmeric powder, folic acid fortified, and NaFeEDTA fortified RW dried turmeric powder samples. This could be due to the reason that the folic acid and NaFeEDTA fortification did not adversely affect the nutritional components of the turmeric powder. However, nutritional analysis could not be carried out for the second day for the case of A as the milk stored under unrefrigerated conditions became a curd and was spoiled. Similar results were obtained for a relevant study reported in the literature. For the case, the nutritional profile of clove-flavoured milk was reported as 58.70 mg/100 g, 46.15 mg/100 g, and 487.10 mg/100g of TPC, TFC, and AA, respectively [50].

For sample B, the sample analysis indicated lower TPC, TFC and AA values for RW dried turmeric powder, folic acid fortified, and NaFeEDTA fortified RW dried turmeric powder with an increase in the number of days (values obtained on Day 2 was lower than the values obtained on Day 1 value). However, the decreasing trend was not apparent for the CC as the curcumin compound has been more stable than TPC, TFC and AA compounds [7]. Reduction in the TPC, TFC, and AA values was apparent for the day 2 sample of B. However, no such analysis could be carried out on day 3 as the golden milk sample was spoiled even after being kept in the refrigerator.

Table 3. A summary of nutritional characteristics of (a) unrefrigerated golden milk products and (b) refrigerated golden milk products

(a)

S. No.

Day

Samples

AA (%)

TPC

(mgGAE/

100 mL)

TFC

(mg quercetin/

100 mL)

CC

(% w/w)

1.

0

Unfortified

24.50±2

876.21±1

784.61±2

4.20±0.1

2.

Folic acid fortified

23.50±1

878.01±3

788.97±1

4.21±0.1

3.

NaFeEDTA fortified

24.10±2

870.87±1

787.49±2

4.19±0.2

(b)

S. No

Day

Samples

AA

(%)

TPC

(mg GAE/

100 mL)

TFC

(mg quercetin/

100 mL)

CC

(% w/w)

1.

1

Unfortified

22.50±1

830.14±3

710.76±2

4.14±0.1

2.

Folic acid fortified

22.80±2

825.51±3

716.87±3

4.16±0.2

3.

NaFeEDTA fortified

22.68±1

829.84±3

717.54±3

4.14±0.1

1.

2

Unfortified

20.90±2

784.29±3

650.71±3

4.10±0.1

2.

Folic acid fortified

20.12±2

786.31±2

646.15±3

4.10±0.2

3.

NaFeEDTA fortified

21.00±2

781.62±2

649.25±2

4.11±0.1

Reviewer 2 Report

Comments and Suggestions for Authors

This manuscript presents a comprehensive investigation into the development of a "golden milk" product using turmeric powder produced via Refractance Window (RW) drying. The study covers a wide range of topics, including the production and fortification of the powder, a detailed storage study to determine shelf-life, sensory analysis using fuzzy logic, and characterization of the final product. However, the manuscript in its current form suffers from several major issues that preclude its acceptance for publication. While the core research question is valuable and the dataset has potential, the manuscript requires a thorough and substantial revision.

  1. The sensory analysis (Section 2.7, 3.4) was conducted to find the optimal concentration of unfortified RW-dried turmeric powder in milk. However, a significant portion of the work, including the shelf-life and storage studies, focuses on powders fortified with folic acid and NaFeEDTA. The manuscript then incorrectly applies the conclusion from the unfortified sensory study to the fortified products. For example, the conclusion states, "The fuzzy logic based sensory analysis...concludes the formulation with 1 g of fortified turmeric powder..." [Line 689-690]. This is factually incorrect, as the fortified powders were explicitly not tested for sensory attributes [Line 208-209]. This invalidates a central theme of the paper and must be addressed, either by performing the missing sensory analysis or by clearly separating the studies and their respective conclusions.
  2. In Section 2.3, the fortification method is described as "dry mixing using a spatula" [Line 146]. For fortifying 100 g of powder with only 20 mg of a fortificant (a 1:5000 ratio), manual mixing with a spatula is grossly inadequate to ensure homogeneity. This method is highly susceptible to error and would result in a non-uniform product, making the subsequent analyses of the fortified powders unreliable. Standard laboratory practice would require a V-blender, planetary mixer, or at least a multi-stage geometric dilution process. The authors must either provide strong evidence of homogeneity.
  3. The discussion of the results, particularly in Sections 3.4 (Sensory Assessment) and 3.6 (Storage Assessment), is highly repetitive and lacks scientific depth. For each sensory attribute (3.4.1-3.4.7), the text follows an identical template: stating the attribute's importance, reporting the highest and lowest scores from the figure, and then moving on. There is no attempt to discuss the interplay between attributes (e.g., how a strong aroma might negatively impact taste perception). Similarly, in the storage study (3.6.1-3.6.14), each subsection presents the data from the table with minimal interpretation beyond stating the trend. For instance, the discussion on TPC reduction [Lines 515-517] mentions enzymatic oxidation but fails to consider the low water activity environment and the potential role of non-enzymatic pathways. The discussion needs to be significantly expanded to provide mechanistic insights and compare findings more critically with the literature.
  4. In the shelf-life assessment (Section 3.3), the critical moisture content (Xc) is stated as 8.60% (db) [Line 274] without any justification. This is a pivotal parameter in the shelf-life calculation. The authors must explain how this value was determined. Was it based on the onset of caking, microbial instability, a significant loss in a key nutritional component, or an unacceptable change in sensory properties? Without this justification, the calculated shelf-life of ~180 days is arbitrary.
  5. The presentation of Table 3 is unacceptable for a scientific publication. The data is split into 14 separate sub-tables (Table 3a through 3n), making it extremely difficult for the reader to follow and compare parameters. All data from the storage study should be consolidated into one or, at most, two well-structured tables.
  6. Figure 3, which presents the sensory data, is also poorly designed. It consists of five separate bar charts (a-e). This data would be far more effectively and efficiently presented in a single grouped bar chart, allowing for direct visual comparison between the different formulations across all sensory attributes.
  7. Throughout the manuscript (e.g., footnotes for Table 2 and all parts of Table 3), standard deviations are reported as a range (e.g., "0.1 – 0.3"). This is not standard scientific practice. The mean and its specific standard deviation (or standard error) should be reported for each individual data point (e.g., 4.00 ± 0.15). This must be corrected for every result presented.
  8. The table must be sorted in the order of appearance, which the author obviously did not do.
  9. In the abstract, it is stated that "The shelf-life of all powders estimated using the GAB model was about 180 days" [Line 23-24]. This is a conceptual error. The GAB model is a sorption isotherm model used to relate water activity to moisture content. It is a tool used within the shelf-life calculation (like the one shown in Eq. 2), but it does not directly yield a shelf-life value. This sentence should be rephrased to be accurate.
  10. The fuzzy logic analysis (Section 3.4.8) is presented in a very confusing manner. The text fails to explain how the numerical similarity values in Table 4 correspond to the qualitative ranks (e.g., "very good," "excellent") mentioned in the text. The link between the methodology (Eq. 4-11), the results (Table 4), and the interpretation is not clear, rendering this section almost incomprehensible to the reader. This section requires a complete rewrite with clear explanations.
  11. While generally understandable, the manuscript contains numerous instances of awkward phrasing, grammatical errors, and suboptimal word choices that detract from its professionalism.
  12. Inconsistent reference formatting. The entire list needs to be carefully checked and formatted according to the journal's specific guidelines.

Comments on the Quality of English Language

Must be improved.

Author Response

Response to reviewer’s comment

Thank you so much for valuable comments and useful insights to improve our manuscript significantly. A detailed account of the responses for all comments is as follows:

Reviewer #2

This manuscript presents a comprehensive investigation into the development of a "golden milk" product using turmeric powder produced via Refractance Window (RW) drying. The study covers a wide range of topics, including the production and fortification of the powder, a detailed storage study to determine shelf-life, sensory analysis using fuzzy logic, and characterization of the final product. However, the manuscript in its current form suffers from several major issues that preclude its acceptance for publication. While the core research question is valuable and the dataset has potential, the manuscript requires a thorough and substantial revision.

Comment 1: The sensory analysis (Section 2.7, 3.4) was conducted to find the optimal concentration of unfortified RW-dried turmeric powder in milk. However, a significant portion of the work, including the shelf-life and storage studies, focuses on powders fortified with folic acid and NaFeEDTA. The manuscript then incorrectly applies the conclusion frosm the unfortified sensory study to the fortified products. For example, the conclusion states, "The fuzzy logic based sensory analysis...concludes the formulation with 1 g of fortified turmeric powder..." [Line 689-690]. This is factually incorrect, as the fortified powders were explicitly not tested for sensory attributes [Line 208-209]. This invalidates a central theme of the paper and must be addressed, either by performing the missing sensory analysis or by clearly separating the studies and their respective conclusions.

Response: We thank the reviewer for the comment and changes have been made accordingly in the revised manuscript.

Line 678 – 693

The present work provides a benchmark for the utilization of RW dried turmeric, folic acid, and iron in the golden milk formulations. Also, the study demonstrates a depth analysis of the shelf life of golden milk formulations, fortified powder mix with various quality parameters. The sorption characteristics confirmed that the GAB model has the highest fit for the fortified and unfortified turmeric samples. The fuzzy logic-based sensory analysis of the golden milk formulations concludes that the formulation with 1 g of unfortified turmeric powder in 100 mL milk is the best formulation. The various quality fractions studied during the storage study did not have significant alterations for the RW dried turmeric powder, folic acid fortified turmeric powder, and NaFeEDTA fortified turmeric powder. These indicated superior shelf life and applicability of the prepared powder mixes. The results indicated that for the fortified turmeric powder mix, the reduction of quality parameters was marginal and was not significant for a duration upto 24 weeks. For the case of the golden milk formulations up to 48 h, there was no significant loss of nutritional parameters. In the future, researchers may explore the applications of fortification of folic acid and NaFeEDTA in various other staple foods with detailed shelf-life studies. This will pave the way for newer horizons in the functional food products.

Comment 2: In Section 2.3, the fortification method is described as "dry mixing using a spatula" [Line 146]. For fortifying 100 g of powder with only 20 mg of a fortificant (a 1:5000 ratio), manual mixing with a spatula is grossly inadequate to ensure homogeneity. This method is highly susceptible to error and would result in a non-uniform product, making the subsequent analyses of the fortified powders unreliable. Standard laboratory practice would require a V-blender, planetary mixer, or at least a multi-stage geometric dilution process. The authors must either provide strong evidence of homogeneity.

Response: We thank the reviewer for the comment and changes have been made accordingly in the revised manuscript.

The work conducted in this article is a follow up of our earlier research article [Talukdar, P., Baruah, K. N., Barman, P. J., Sharma, S., & Uppaluri, R. V. (2025). Development and Characterization of Refractance Window-Dried Curcuma longa Powder Fortified with NaFeEDTA and Folic Acid: A Study on Thermal, Morphological, and In Vitro Bio Accessibility Properties. Foods14(4), 658.. The homogenization of the mixture was ensured by following the process of dry mixing being reported in another published article [Karn, S. K., Chavasit, V., Kongkachuichai, R. & Tangsuphoom, N. Shelf stability, sensory qualities, and bioavailability of iron-fortified Nepalese curry powder. Food Nutrition. Bulletin. 32, 13–22 (2011)]. Since the fortified turmeric powder was eventually mixed in milk to obtained golden milk, its created a homogenous mixture. Therefore, no expensive mixing method was to be followed as it is not required to achieve. Also, the conducted research was anticipated to be useful for household communities.

Comment 3: The discussion of the results, particularly in Sections 3.4 (Sensory Assessment) and 3.6 (Storage Assessment), is highly repetitive and lacks scientific depth. For each sensory attribute (3.4.1-3.4.7), the text follows an identical template: stating the attribute's importance, reporting the highest and lowest scores from the figure, and then moving on. There is no attempt to discuss the interplay between attributes (e.g., how a strong aroma might negatively impact taste perception). Similarly, in the storage study (3.6.1-3.6.14), each subsection presents the data from the table with minimal interpretation beyond stating the trend. For instance, the discussion on TPC reduction [Lines 515-517] mentions enzymatic oxidation but fails to consider the low water activity environment and the potential role of non-enzymatic pathways. The discussion needs to be significantly expanded to provide mechanistic insights and compare findings more critically with the literature.

Response: We thank the reviewer for the comment and changes have been made accordingly in the revised manuscript.

Line 379 – 419

3.4. Sensory Assessment of Powder Product

3.4.1 Hedonic scale based sensory analysis

As a simple marketing rule, if consumers do not like the appearance, flavour, or texture of a food product, they will not buy it. Therefore, the overall sensory experience of a product is crucial for its commercial success. Specific protocols and methods have been developed to estimate and quantify consumers' sensory experiences. Accordingly, the risk associated with the non-acceptability of a food product can be reduced through the scientific relevance in terms of the sensory descriptive analysis or sensory descriptive evaluation. Appropriate sensory evaluation allows a very useful understanding of the key attributes that assist in the commercial success of food products [47].

The RW dried turmeric powder has been evaluated for its sensory characteristics. However, the folic acid fortified and NaFeEDTA fortified turmeric powder samples were not analysed for their sensory characteristics. This is due to the non-food grade status of the deployed folic acid and NaFeEDTA fortificants. The sensory analysis was conducted with a panel of judges and in terms of the scores provided for colour and appearance, taste, aroma, mouthfeel, aftertaste, consistency, and overall acceptability [48].

The colour and appearance of a product are a prime entity for its formal acceptance or rejection during sensory analysis. The aesthetic quality of a food product is dominantly influenced by its colour and thereby provides visual inputs for flavour identification. After colour and appearance, aroma has significance in the sensory evaluation of a sample and thereby critically influences the formal acceptance or rejection of a sample by a consumer. The smell or aroma of a product influences the olfactory glands of consumers and accordingly enhances the desire of the consumers to taste a product. Thus, a product with a good aroma attracts a consumer to taste it, and a bad or strong aroma hinders the formal acceptance of the food product.  Despite having good appearance and colour properties, a food product is usually accepted for its good taste. The aftertaste is an attribute that lingers in the palate of the mouth after a sample's ingestion into the mouth. The aftertaste of the product is an important sensory attribute and reflects upon the acceptability of the product by the consumers. The consistency and mouthfeel of a product is an important sensory attribute to infer upon its degree of acceptability and product's acceptability. However, the overall acceptability parameter has the final say in the acceptability of a product by the consumer.[47,48]

The sensory analysis was performed for 100 ml milk mixed with 0g, 0.5, 1g, 1.5g and 2 g of RW dried turmeric powder. Twenty-five semi trained panellists had evaluated the samples for the sensory attributes - colour and appearance, aroma, taste, aftertaste, consistency, mouthfeel and overall acceptability – and with the 9 score hedonic scale (Fig 3). At lower concentrations, i.e., 0 g and 0.5 g, the sensory scores for colour were liked marginally. However, other attributes were moderately liked. The sensory scores for 1 g concentration were highest between 8 to 9 (Like very much and like moderately) for all the attributes. At 2g, the scores for mouthfeel, aftertaste, and overall acceptability suggested that the sample is disliked by panellists.

Line 490 – 676

3.6.1. Moisture Content (MC)

The MC is one of the most important factors in the due course of the evaluation of the quality and stability of the RW dried turmeric powder, folic acid fortified RW dried turmeric powder, and NaFeEDTA fortified RW dried turmeric powder samples. Thus, low MC of the samples during storage affirms a decelerated and reduced rate of several degradation reactions and associated microbial growth [3]. It is well known that the powder samples with high MC are susceptible to quality deterioration at higher temperatures. This is due to the hydrolysis of oil and phospholipids followed by an increase in the sample acidity[52]. Various associated studies for the storage of powdered food products confirmed that the optimal MC of dried powders shall be in the range of 4 – 8 % for good storage stability[53], [54]. During the storage study, the MC of all samples gradually increased with time (Table 4a). Thereby, the powders may undergo transformations (caking) with time, and may also get modified due to pertinent long-term effects of the environmental and mechanical conditions[17], [55]. Since moisture content increase with time occurs due to the migration of water vapour from the storage environment into the packaging material, the chances of caking also increases with time. However, for all tested samples, no caking was observed till 180 days of storage under ambient conditions[52].

3.6.2. Anti-oxidant Activity

Using the DPPH method, the initial AA values of the RW dried turmeric powder, folic acid fortified RW dried turmeric powder, and NaFeEDTA fortified RW dried turmeric powder samples have been obtained without any variation and as 90.00 – 89.91 % (Table 4 b). After accelerated storage, the AA was reduced to about 84.80 – 84.00 %. The AA values of RW dried turmeric powder, folic acid fortified RW dried turmeric powder, and NaFeEDTA fortified RW dried turmeric powder samples did not reduce significantly with time. Hence, the pertinent losses have been insignificant. The storage temperature did not alter the AA reduction trend with time. This is probably due to the AA attribute not being related to a single or several similar compounds but to a class of compounds that exhibited synergy in terms of their respective antioxidant activities[17], [56]. Thus, the AA showed greater interaction of various constituents leading to stability of AA with storage time [55].

3.6.3. Total Phenolic Content

The variation in TPC with time for the tested samples has been presented in Table 4 (c). A marginal reduction in TPC was apparent for RW dried turmeric powder, folic acid fortified RW dried turmeric powder, and NaFeEDTA fortified RW dried turmeric powder samples during the storage period of 180 days. The initial total phenolic content of unfortified and fortified turmeric powders was 190.00 – 189.01 mg GAE/100 g. After 180 days of storage, the TPC was reduced to 167.01 – 165.00 mg GAE/100 g. Such a reduction in the phenolic content was due to the oxidation of phenolic compounds along with the activation of oxidative enzymes such as polyphenoloxidase [57]. With time, there is some increase in moisture content, and higher moisture content conveys more molecular mobility which may lead to higher degradation rate. Similar results have been reported for apple peel powder samples at 4, 10, and 25 oC [58], papaya powder samples at 30 oC and for freeze-dried strawberry puree[17], [59].

3.6.4. Total Flavonoid Content

The initial TFC of RW dried turmeric powder, folic acid fortified RW dried turmeric powder, and NaFeEDTA fortified RW dried turmeric powder samples were 161.02 – 159.10 mg/100 g. During accelerated storage, the TFC of the samples was reduced to 140.00 – 138.00 mg quercetin/100 g (Table 4d). Thus, the trends have been apparently similar to those observed for the TPC. Such synergy is caused by the fact that the flavonoids are the major phenolic compounds. Henceforth, the TFC correlated and exhibited a similar pattern to reported TPC alteration during the storage period of a relevant sample [57]. Also, at room temperature, TFC may oxidize more rapidly, and this results in decreased TFC retention over storage time [60]. In a relevant literature, the authors reported that the microencapsulated kenaf seed oil under accelerated storage conditions indicated a reduced TFC in the course of the storage[61]. Similarly, authors reported a rapid degradation of TFC under higher temperature conditions[57], [62].

3.6.5. Curcumin Content

The yellow colour in the turmeric is due to the polyphenolic constituent curcumin that possesses lipophilic characteristics. In the fresh turmeric sample, the CC was 0.73 % w/w. For RW dried turmeric powder, folic acid fortified RW dried turmeric powder, and NaFeEDTA fortified RW dried turmeric powder samples, the curcumin content was 4.84 – 4.80 % w/w (Table 4e). After 180 days' storage period, the curcumin content alterations have been insignificant (4.68 – 4.65 % w/w). Another related study reported the stability of curcumin during a 28-day storage period for curcumin fortified yogurt [63]. However, curcumin is prone to photodegradation under visible light and to oxidative degradation upon exposure to the oxygen. Moreover, long-term storage leads to a gradual loss of curcumin content, and in even well-controlled environments [61].

3.6.6. Colour Indices

For RW dried turmeric powder, folic acid fortified RW dried turmeric powder, and NaFeEDTA fortified RW dried turmeric powder samples, the variation of colour parameters in the course of the storage period has been summarized in Table 4 (f). The results confirmed that the L value and whiteness attribute of the tested samples remain almost unaltered during storage. However, a marginal reduction in a and b values can be especially observed during the storage. Thus, storage period marginally influenced the reduction of a and b values.  In this regard, it is known that while b values indicate yellowness (+) or blueness (−), the a values refer to redness (+) or greenness (-) of the tested samples. The colour of food products during storage is affected by various factors, including packaging material, storage temperature, sugar and protein composition, water activity, and storage time. Also, the curcumin content critically contributes to the turmeric colour. This is due to the pigmented curcumin compounds that impart its yellow color. Henceforth, good curcumin stability in dried turmeric led to good colour stability of the product [63].

3.6.7. Folic Acid

The folic acid content variation in the folic acid fortified turmeric samples during accelerated storage conditions has been summarized in Table 4 (g).  For a storage time variation up to 180 days, the total folic acid content reduced from 20.00 to 18.74 mg/100 g sample of turmeric powder. Such a marginal reduction in folic acid content has been due to the stable nature of the folic acid used for fortification [64]. Another study reported similar results for the stability of folic acid after fortification in blackberry powder [65]. Also, the physicochemical characteristics of the RW dried turmeric powder did not alter significantly due to folic acid fortification.

3.6.8. Iron Content

For the NaFeEDTA fortified RW dried turmeric sample, the variation in iron content during storage conditions has been summarized in Table 4 (h). With time, an insignificant reduction in iron content has been apparent. The iron content ranged from 20.00 to 19.41 mg/100 g sample.  Such an insignificant reduction has been due to the stability of the turmeric powder and NaFeEDTA powder. This aspect did not result in a significant interaction. Also, the inorganic and stable nature of the iron powder fostered a minimum loss of iron content with time in the NaFeEDTA fortified turmeric sample [66].

3.6.9. Bulk Density

The bulk density of the turmeric powder samples increased with storage time (Table 4i). This was due to the moisture gain of the powder samples. A similar enhancement in the bulk densities with increasing MC during storage has been reported in previous studies for the mango milk powder system[67].  An increase in bulk density may also be attributed to enhanced cohesiveness between powder particles. This is due to the enhanced absorption of moisture during the storage period. Also, prolonged storage leads to particle settling and compaction. This inturn increases bulk density. However, the bulk density was stable for folic acid fortified spray-dried blackberry in a related prior art [65].

3.6.10. Solubility

The alterations in the solubility of RW dried turmeric powder, folic acid fortified RW dried turmeric powder, and NaFeEDTA fortified RW dried turmeric powder samples during storage at various RH levels have been presented in Table 4 (j). After RW drying, the turmeric powders possessed a solubility of 30 – 29.00 and almost underwent complete dissolution in water. However, with time, the solubility of all tested samples did not alter up to the 9th week. Thereafter, the parameter reduced marginally to 25 – 26.00 at the end of the storage period (week 24). These findings confirmed that the solubility of all samples was influenced by the relative humidity conditions of the storage environment, and a higher RH level contributed to marginal solubility loss[3]. With an increase in storage time, there was a marginal enhancement in the moisture content of the sample. This eventually lead to a loss in the solubility. Such insignificant solubility loss was due to a minor enhancement in moisture content that has an adverse influence on the solubility[68]. Another possible reason could be the protein-carbohydrate interaction (Maillard browning), which reduces solubility with time [69].

3.6.11. Swelling Power

The swelling power of RW dried turmeric powder, folic acid fortified RW dried turmeric powder, and NaFeEDTA fortified RW dried turmeric powder samples varied marginally during the duration of the storage period (Table 4k). During the initial storage period, the swelling power remained fairly constant at 1.8 g/g (up to week 9). However, after the 9th week, a marginal reduction in swelling power was noticed. Such parametric reduction did not significantly increase with time during the later stages of the storage. Thus, for the maximum storage period of 180 days, the reduction in swelling power was not significant (value of 1.5 g/g). The lower reduction in swelling power with storage period has been due to the stable nature of the turmeric powder. Since all tested samples did not significantly absorb moisture content, their stability remained intact in terms of good swelling power during the total time period of the storage study [68], [69].

3.6.12. Water Binding Capacity

The water binding capacity of RW dried turmeric powder, folic acid fortified RW dried turmeric powder, and NaFeEDTA fortified RW dried turmeric powder samples has been summarized in Table 4 (l). Long-term storage leads to gradual structural changes (e.g., crystallization of amorphous sugars), and reduces the WHC. The table infers that with storage time, a marginal reduction in water binding capacity occurred. However, the reduction has been insignificant. Up to week 9, the water binding capacity remained unchanged (66 %). However, after week 17, the reduction was apparent but marginal. This may be due to the stable nature of the tested samples that absorbed less moisture during storage. Hence, high water binding capacity could be retained even at longer storage period conditions [69], [70].

3.6.13. Dispersion Time

The reconstitution properties of powdered food products ensured their acceptability in the consumer market. Dispersion has been one of the most important reconstitution properties. It is defined as the pace at which the powder dissolves upon reconstitution in water. In other words, any powder material can be inferred to be of best quality for the scenario in which the sample’s particles get immediately dissolved in water and without the creation of the lumps. The dispersion time of the RW dried turmeric powder, folic acid fortified RW dried turmeric powder, and NaFeEDTA fortified RW dried turmeric powder samples varied between 20 and 17 s (Table 4 m). The reduced dispersibility over longer storage periods could be due to the powder's rising moisture content, which possibly enhances particle cohesiveness and henceforth facilitates lump formation during water dissolution [44]. Also, insignificant differences have been recorded in the dispersibility of RW dried turmeric powder, folic acid fortified RW dried turmeric powder, and NaFeEDTA fortified RW dried turmeric powder samples after the second week. Thereby, good confidence levels have been ensured, and the findings have been in corroboration with relevant prior art [52], [69], [70].

3.6.14. Hygroscopicity

Table 4 (n) summarizes the findings associated with the hygroscopicity of RW dried turmeric powder, folic acid fortified RW dried turmeric powder, and NaFeEDTA fortified RW dried turmeric powder samples being subjected to accelerated storage conditions up to 180 days. The initial hygroscopicity of RW dried turmeric powder, folic acid fortified RW dried turmeric powder, and NaFeEDTA fortified RW dried turmeric powder samples were 8.80 – 8.70 g/100g. Thereafter, the parameter value was reduced to 8.10 – 8.00 g/100 g in the course of accelerated storage conditions. While the powder exhibited minimal moisture uptake initially, prolonged storage resulted in progressive moisture absorption [15], [69]. This has probably been due to the enhanced moisture content in the powder that eventually led to further reduction in the absorbed water [70].

Comment 4: In the shelf-life assessment (Section 3.3), the critical moisture content (Xc) is stated as 8.60% (db) [Line 274] without any justification. This is a pivotal parameter in the shelf-life calculation. The authors must explain how this value was determined. Was it based on the onset of caking, microbial instability, a significant loss in a key nutritional component, or an unacceptable change in sensory properties? Without this justification, the calculated shelf-life of ~180 days is arbitrary.

Response: We thank the reviewer for the comment and changes have been made accordingly in the revised manuscript.

Line 355 – 378

Using Eq. 2, the shelf-life parameters of RW dried turmeric powder, folic acid fortified, and NaFeEDTA fortified RW dried turmeric powder samples packed in zipper pouches (38 ± 1 oC) were determined. The water vapour permeability of zipper pouches was calculated using Eq. 3 and was found to be 1.12×10-7 kg/m2dayPa, respectively. The initial moisture content (Xi) of RW dried turmeric powder, folic acid fortified, and NaFeEDTA fortified RW dried turmeric powder samples was the same for all cases at 4.27 % (db), respectively. After 180 days of storage period, the final moisture content of packed in RW dried turmeric powder, folic acid fortified RW dried turmeric and iron fortified RW dried turmeric powder samples were 8.60, 8.70 and 8.50 % (db), respectively. Thus, it is apparent that the moisture content of the powder sample gradually increased with storage time. Also, at this stage the sample showed caking tendency. [34]. Therefore, the critical moisture content (Xc) was taken at this stage as 8.60 % (db) for all cases.

At a storage temperature of 38 oC, and for RW dried turmeric powder, folic acid fortified RW dried turmeric, and NaFeEDTA fortified RW dried turmeric powder samples, the water activity values corresponding to critical moisture content were 0.49, 0.49, and 0.48, respectively. Considering the zipper pouch surface area as 0.014 m2 for all cases, the total solid weight (Ws) is 0.01918, 0.01938, and 0.01938 kg for RW dried turmeric powder, folic acid fortified RW dried turmeric, and NaFeEDTA fortified RW dried turmeric powder samples, respectively. The saturated vapour pressure of 6980.5 Pa, has been used to determine the shelf life of the samples[45]. Corresponding values were 184, 187, and 183 days for RW dried turmeric powder, folic acid fortified RW dried turmeric, and NaFeEDTA fortified RW dried turmeric powder samples, respectively, in the zipper pouch system. In a relevant study, the authors reported a shelf life of 102 days for milk millet powders, which is comparable to the results obtained in this work[46].

Comment 5: The presentation of Table 3 is unacceptable for a scientific publication. The data is split into 14 separate sub-tables (Table 3a through 3n), making it extremely difficult for the reader to follow and compare parameters. All data from the storage study should be consolidated into one or, at most, two well-structured tables.

Response: We thank the reviewer for the comment and changes have been made accordingly in the revised manuscript.

Table 4.: Time-dependent data of stored refractance window dried turmeric powder products.

(a)    Moisture Content

S. No.

Sample

0 weeks

9 weeks

17 weeks

24 weeks

1.

Unfortified

4.00±0.1

4.90±0.1

6.10±0.1

7.81±0.2

2.

Folic acid fortified

4.00±0.2

4.95±0.2

6.10±0.1

7.80±0.2

3.

NaFeEDTA fortified

4.00±0.3

4.91±0.2

6.30±0.1

7.83±0.3

(b)   Anti-oxidant Activity

S. No.

Sample

0 weeks

9 weeks

17 weeks

24 weeks

1.

Unfortified

90.00±0.5

88.60±0.6

85.40±0.3

84.80±0.3

2.

Folic acid fortified

89.91±0.6

88.70±0.2

85.90±0.5

84.00±0.4

3.

NaFeEDTA fortified

89.97±0.6

87.40±0.5

85.10±0.2

84.50±0.5

(c)Total Phenolic Content

S. No.

Sample

0 weeks

9 weeks

17 weeks

24 weeks

1.

Unfortified

189.76±2

180.00±3

175.23±2

167.01±2

2.

Folic acid fortified

190.00±2

179.12±2

176.31±2

165.00±2

3.

NaFeEDTA fortified

189.01±4

181.15±2

173.47±2

166.25±2

(d) Total Flavonoid Content

S. No.

Sample

0 weeks

9 weeks

17 weeks

24 weeks

1.

Unfortified

160.00±4

153.00±5

145.00±2

139.00±4

2.

Folic acid fortified

159.10±3

152.00±5

147.00±2

138.00±3

3.

NaFeEDTA fortified

161.02±2

151.00±4

146.00±1

140.00±3

(e) Curcumin Content

S. No.

Sample

0 weeks

9 weeks

17 weeks

24 weeks

1.

Unfortified

4.84±0.02

4.75±0.03

4.70±0.02

4.65±0.01

2.

Folic acid fortified

4.80±0.01

4.76±0.03

4.71±0.02

4.67±0.01

3.

NaFeEDTA fortified

4.83±0.02

4.77±0.02

4.72±0.02

4.68±0.02

(f) Colour Indices

S. No.

Sample

0 weeks

9 weeks

17 weeks

24 weeks

L

a

b

L

a

b

L

a

b

L

a

b

1.

Unfortified

56±2

31±1

62±3

55±2

29±1

60±2

53±1

26±3

57±2

51±2

23±1

53±3

2.

Folic acid fortified

55±2

30±2

61±3

54±2

29±1

60±2

52±3

26±3

56±1

51±2

23±1

53±2

3.

NaFeEDTA fortified

56±2

30±1

61±1

54±1

29±1

60±1

53±1

26±3

57±1

52±2

23±1

53±2

(g) Folic acid

S.No.

Sample

0 weeks

9 weeks

17 weeks

24 weeks

1.

Folic acid fortified

20.00±0.5

19.65±0.5

19.01±1

18.74 ±1

(h) Iron content

S. No.

Sample

0 weeks

9 weeks

17 weeks

24 weeks

1.

NaFeEDTA fortified

20.00±0.1

19.91±0.2

19.72±0.2

19.41±0.3

(i)    Bulk density

S. No.

Sample

0 weeks

9 weeks

17 weeks

24 weeks

1.

Unfortified

0.62±0.01

0.64±0.04

0.67±0.04

0.69±0.01

2.

Folic acid fortified

0.65±0.01

0.66±0.05

0.68±0.02

0.70±0.03

3.

NaFeEDTA fortified

0.64±0.03

0.65±0.05

0.67±0.03

0.69±0.03

(j)    Solubility

S. No.

Sample

0 weeks

9 weeks

17 weeks

24 weeks

1.

Unfortified

29.00±1

29.00±2

27.00±1

26.00±3

2.

Folic acid fortified

30.00±3

30.00±2

28.00±1

27.00±3

3.

NaFeEDTA fortified

28.00±1

28.00±3

26.00±1

25.00±2

(k)   Swelling power

S. No.

Sample

0 weeks

9 weeks

17 weeks

24 weeks

1.

Unfortified

1.80±0.1

1.80±0.2

1.70±0.3

1.50±0.1

2.

Folic acid fortified

2.00±0.1

2.00±0.2

1.90±0.2

1.70±0.3

3.

NaFeEDTA fortified

1.90±0.2

1.90±0.2

1.80±0.3

1.60±0.3

(l)    Water holding capacity

S. No.

Sample

0 weeks

9 weeks

17 weeks

24 weeks

1.

Unfortified

66.00±1

66.00±2

64.00±1

61.00±2

2.

Folic acid fortified

65.00±2

65.00±2

63.00±1

59.00±1

3.

NaFeEDTA fortified

67.00±2

67.00±2

65.00±1

60.00±1

(m)  Dispersion time

S. No.

Sample

0 weeks

9 weeks

17 weeks

24 weeks

1.

Unfortified

20±2

22±2

26±1

30±2

2.

Folic acid fortified

17±2

21±2

25±1

29±2

3.

NaFeEDTA fortified

19±2

23±2

28±1

31±2

(n)   Hygroscopicity

S. No.

Sample

0 weeks

9 weeks

17 weeks

24 weeks

1.

Unfortified

8.70±0.1

8.60±0.1

8.40±0.2

8.10±0.1

2.

Folic acid fortified

8.80±0.2

8.60±0.1

8.50±0.2

8.00±0.2

3.

NaFeEDTA fortified

8.50±0.2

8.40±0.2

8.30±0.1

8.00±0.2

Comment 6: Figure 3, which presents the sensory data, is also poorly designed. It consists of five separate bar charts (a-e). This data would be far more effectively and efficiently presented in a single grouped bar chart, allowing for direct visual comparison between the different formulations across all sensory attributes.

Response: We thank the reviewer for the comment and changes have been made accordingly in the revised manuscript.

Figure 3. Radar chart depicting sensory characteristics of refractance window dried turmeric powder-based milk products for varied turmeric constitution cases

Comment 7: Throughout the manuscript (e.g., footnotes for Table 2 and all parts of Table 3), standard deviations are reported as a range (e.g., "0.1 – 0.3"). This is not standard scientific practice. The mean and its specific standard deviation (or standard error) should be reported for each individual data point (e.g., 4.00 ± 0.15). This must be corrected for every result presented.

Response: We thank the reviewer for the comment and changes have been made accordingly in the revised manuscript.

The modified Table is shown at comment 5.

Comment 8: The table must be sorted in the order of appearance, which the author obviously did not do.

Response: We thank the reviewer for the comment and changes have been made accordingly in the revised manuscript.

Comment 9: In the abstract, it is stated that "The shelf-life of all powders estimated using the GAB model was about 180 days" [Line 23-24]. This is a conceptual error. The GAB model is a sorption isotherm model used to relate water activity to moisture content. It is a tool used within the shelf-life calculation (like the one shown in Eq. 2), but it does not directly yield a shelf-life value. This sentence should be rephrased to be accurate.

Response: We thank the reviewer for the comment and changes have been made accordingly in the revised manuscript.

Line: 27 - 30

The results inferred upon a healthy shelf life of 184, 187, and 183 days for RW dried turmeric powder, folic acid fortified, and NaFeEDTA fortified RW dried turmeric powder samples, respectively, in the zipper pouch system.

Comment 10: The fuzzy logic analysis (Section 3.4.8) is presented in a very confusing manner. The text fails to explain how the numerical similarity values in Table 4 correspond to the qualitative ranks (e.g., "very good," "excellent") mentioned in the text. The link between the methodology (Eq. 4-11), the results (Table 4), and the interpretation is not clear, rendering this section almost incomprehensible to the reader. This section requires a complete rewrite with clear explanations.

Response: We thank the reviewer for the comment and changes have been made accordingly in the revised manuscript.

Line: 421 - 441

The conducted study applied fuzzy logic for the sensory attributes-based evaluation of the sensitive influence of the turmeric powder concentration. The similarity values were reflective of the ranks being obtained as – not satisfactory - 0.00 to 0.40, satisfactory – 0.40 – 0.60, good -0.60 – 0.80, very good – 0.80 - 0.85, and excellent – 0.85 – 1.00. The quality attributes ranking from fuzzy logic has been summarized in Table 2. The similarity value was calculated based on fuzzy logic method and the assigned membership for a concentration of 1.5 g of the turmeric powder has been very good for the color attribute (0.825). Higher similarity values indicated the superiority of the sensory scores for that particular sample. The highest similarity values for all the other parameters except for colour were found for 1g turmeric powder concentration sample. For aroma, while the similarity values indicated very good rank, the parameter values for all other attributes namely taste, mouthfeel, aftertaste, consistency, and overall acceptability were ranked to be excellent. As the sample turmeric powder concentration increased above 1.0 g, the attributes rank reduced to to as satisfactory and as fair. Thus, the fuzzy logic aided in the quantification of the subjective sensory data, and renders the easy analysis and comparison of the effects of different concentrations on the attributes. No prior studies are available for the fuzzy logic-based sensory optimization of turmeric powder products. In a recent study, fuzzy logic analysis conveyed that the probiotic γ-aminobutyric acid has better sensory characteristics with respect to the non-probiotic bar [49]. Similar results were reported for the fuzzy logic basis sensory evaluation of the carrot juice. For the case, the thermosonication improved the taste and colour of carrot juice [18].

Comment 11: While generally understandable, the manuscript contains numerous instances of awkward phrasing, grammatical errors, and suboptimal word choices that detract from its professionalism.

Response: We thank the reviewer for the comment and changes have been made accordingly in the revised manuscript.

Comment 12: Inconsistent reference formatting. The entire list needs to be carefully checked and formatted according to the journal's specific guidelines.

Response: We thank the reviewer for the comment and changes have been made accordingly in the revised manuscript.

Round 2

Reviewer 2 Report

Comments and Suggestions for Authors

Can be accepted.